TOPICAL REVIEW

# Cells and ionic conductances contributing to spontaneous activity in bladder and urethral smooth muscle

Bernard T. Drumm [ID], Neha Gupta [ID], Alexandru Mircea and Caoimhin S. Griffin

*Smooth Muscle Research Centre, Department of Life & Health Science, Dundalk Institute of Technology, Dundalk, Ireland*

Handling Editors: Laura Bennet and T. Alexander Quinn

The peer review history is available in the Supporting Information section of this article (https://doi.org/10.1113/JP284744#support-information-section).

**Abstract figure legend** Cells and conductances contributing to spontaneous activity in the lower urinary tract. Bladder and urethra exhibit spontaneous contractions at both cellular and tissue levels. Both detrusor and urethral smooth muscle cells display activity that is regular and rhythmic. Distinct populations of interstitial cells exist in muscle layers of both organs, with platelet derived growth factor receptor-$\alpha^+$ cells present in detrusor, and Kit$^+$ cells (interstitial cell of Cajal-like cells) in urethra. These cells may influence activity of detrusor and urethral smooth muscle, respectively. Detrusor and urethral smooth muscle cells rely on varying complements of ion channels to regulate spontaneous activity. In bladder, small and large conductance potassium channels (SK3/BK) and voltage-dependent calcium channels (Cav1.2) are consistently found to be important. In urethra, there is disparity among species and investigators as to the importance of Cav1.2, calcium activated chloride (Ano1) channels and Orai calcium channels. This review summarizes the current thoughts of the field on these similarities and discrepancies.

**Abstract**   Smooth muscle organs of the lower urinary tract comprise the bladder detrusor and urethral wall, which have a reciprocal contractile relationship during urine storage and micturition. As the bladder fills with urine, detrusor smooth muscle cells (DSMCs) remain relaxed to accommodate increases in intravesical pressure while urethral smooth muscle cells (USMCs) sustain tone to occlude the urethral orifice, preventing leakage. While neither organ displays coordinated regular contractions as occurs in small intestine, lymphatics or renal pelvis, they do exhibit patterns of rhythmicity at cellular and tissue levels. In rabbit and guinea-pig urethra, electrical slow waves are recorded from USMCs. This activity is linked to cells expressing vimentin, c-kit and $Ca^{2+}$-activated $Cl^-$ channels, like interstitial cells of Cajal in the gastro-intestinal tract. In mouse, USMCs are rhythmically active (firing propagating $Ca^{2+}$ waves linked to contraction), and this cellular rhythmicity is asynchronous across tissues and summates to form tone. Experiments in mice have failed to demonstrate a voltage-dependent mechanism for regulating this rhythmicity or contractions *in vitro*, suggesting that urethral tone results from an intrinsic ability of USMCs to 'pace' their own $Ca^{2+}$ mobilization pathways required for contraction. DSMCs exhibit spontaneous transient contractions, increases in intracellular $Ca^{2+}$ and action potentials. Consistent across numerous species, including humans, this activity relies on voltage-dependent $Ca^{2+}$ influx in DSMCs. While interstitial cells are present in the bladder, they do not 'pace' the organ in an excitatory manner. Instead, specialized cells (PDGFR$\alpha^+$ interstitial cells) may 'negatively pace' DSMCs to prevent bladder overexcitability.

(Received 1 December 2023; accepted after revision 2 September 2024; first published online 26 September 2024)

**Corresponding author** B. T. Drumm: Smooth Muscle Research Centre, Department of Life & Health Science, Dundalk Institute of Technology, Dundalk, Co. Louth, Ireland.     Email: bernard.drumm@dkit.ie

## Introduction

Smooth muscle cells (SMCs) of bladder and urethra display reciprocal contractile patterns vital for urine storage and voiding (Brading, 1999). Bladder SMCs (detrusor SMCs; DSMCs) remain relatively relaxed as the organ fills with urine, accommodating large increases in volume without corresponding increases in intra-vesical pressure (Andersson & Arner, 2004). During storage, urethral smooth muscle (USM) tonically contracts, preventing urine leakage. At micturition onset, urethral musculature relaxes in response to neural release of nitric oxide (NO) (Persson & Andersson, 1992; Waldeck et al., 1998), while DSMCs contract from cholinergic stimulation (de Groat et al., 2001; Hill, 2015).

Contraction and relaxation of bladder and urethra are well coordinated. Urethral pressure must be greater than bladder pressure to avoid urine leakage (Delancey & Ashton-miller, 2004). If intraurethral pressure falls due to inability of urethral muscle to contract, incontinence can result (Drumm et al., 2021). Similarly, hyperexcitable DSMCs can result in increased urination frequency, urgency or incomplete bladder emptying (Andersson & Wein, 2004; Delancey & Ashton-miller, 2004; Keane & O'Sullivan, 2000).

During the filling phase of the micturition cycle, DSMCs remain relatively relaxed while USM is tonically contracted. However, underlying regular spontaneous activity can be recorded from both tissues during this phase. Both the bladder and USM exhibit rhythmic and spontaneous oscillations in either membrane potential

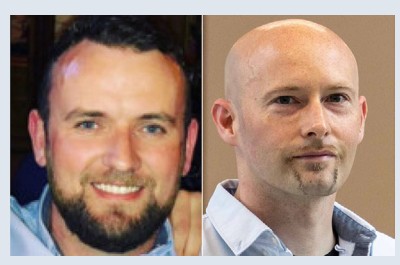

Both authors are PhD graduates of the Smooth Muscle Research Centre (SMRC) at Dundalk Institute of Technology, Ireland. The work of **Caoimhin Griffin** has focused on the ionic regulation of bladder detrusor smooth muscle, which he continued during a postdoctoral fellowship at University of Nevada, Reno. Returning to Dundalk in 2021 as a lecturer and SMRC principal investigator, he now studies how Orai-STIM interactions affect spontaneous activity of bladder tissue. The research of **Bernard Drumm** focuses on calcium signalling mechanisms in smooth muscle and interstitial cells of the urethra and gastro-intestinal tract. After PhD training, he spent 7 years in the Department of Kenton Sanders at University of Nevada Reno, returning to Ireland in 2020. Now a lecturer and investigator in the Dundalk SMRC, he studies the origin of myogenic tone in urethral muscles, with an emphasis on how tone can be modulated to treat incontinence.

or intracellular $Ca^{2+}$ levels (or both) which cause spontaneous regular contractions to varying degrees (Fig. 1). This review summarizes how spontaneous contractions and relaxations of smooth muscle layers of these lower urinary tract organs are generated and how they contribute to continence during the filling phase of micturition. We explore underlying ionic and $Ca^{2+}$ signalling pathways responsible for this activity and contributions of non-smooth-muscle cell-types.

## Urethra anatomy and function

The urethra forms a passage for transfer of urine from the bladder to the exterior (Fletcher & Bradley, 1978) and is a sphincter preventing bladder leakage (Keane & O'Sullivan, 2000). The urethra comprises an internal sphincter of urethral SMCs (USMCs), of circular (outer) and longitudinal (inner) layers (Pel et al., 2006).

There is an external urethral sphincter (rhabdosphincter) of striated muscles (Creed and Van der Werf, 2001). Contractions of rhabdosphincter are under voluntary control, and contribute a 'guarding reflex' to maintain continence on occasions of sudden increases in abdominal and intravesical bladder pressure (sneezing, coughing and laughing) (Fowler et al., 2008). It has historically been assumed that skeletal muscle is the most clinically relevant element of urethral musculature contributing to tone, yet there is increasing clinical awareness that contractions of USM is the dominant input to urethral closure pressure at rest and during bladder filling (Jankowski et al., 2006; Venema et al., 2023). Urethral closure pressure is highest at regions of where USM is most abundant (Awad & Downie, 1976; Bridgewater et al., 1993; Conte et al., 1991; Greenland et al., 1996) and neuromuscular blockade of the rhabdosphincter does not result in urine leakage at rest (Conte et al., 1991; Greenland et al., 1996), suggesting the majority of urethral tone contributing to continence

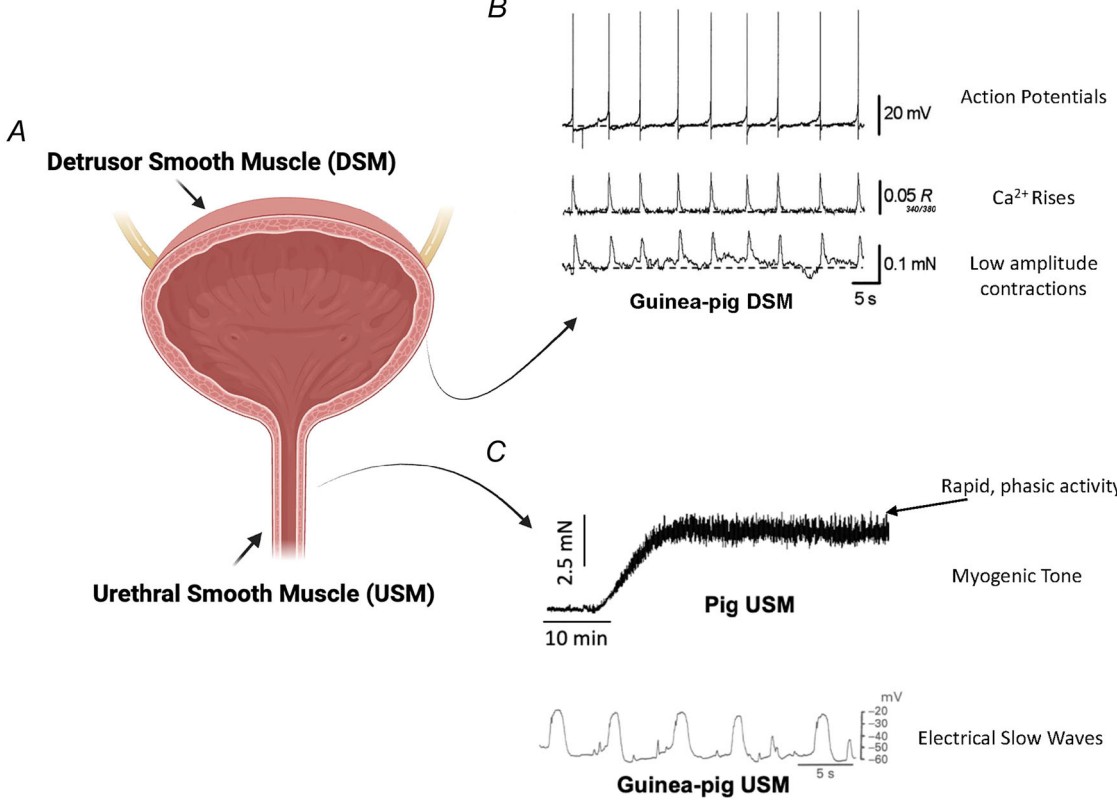

**Figure 1. Spontaneous activity in bladder and urethral smooth muscle**
*A*, the smooth muscle walls of the lower urinary tract consist of the detrusor smooth muscle (DSM) of the bladder and the urethral smooth muscle (USM) of the internal urethral sphincter. *B*, during bladder filling, the DSM displays spontaneous phasic contractile activity, manifesting as low amplitude, non-voiding contractions which are generated from action potential firing and rises in intracellular DSM $Ca^{2+}$ (from Hashitani, Brading et al., 2004). *C*, also during bladder filling, USM generates a sustained myogenic tone to prevent urine leakage from the bladder. In certain species, rapid, low amplitude phasic activity may be superimposed upon tonic contractions (upper panel, from Rembetski et al., 2020). Urethral tissues can display electrical rhythmicity, as regular depolarizations of membrane potential, termed 'slow waves' (lower panel, from Hashitani et al., 1996).

is from USM contraction. The urethral vasculature may also contribute to tone and urethral resistance pressure by applying a passive turgor pressure initiated by altering blood flow to the mucosal layer (Greenland & Brading, 1997; Hashitani, Mitsui et al., 2024).

USM is contracted by neural inputs, predominantly from sympathetic activation of $\alpha$1-adrenoceptors (Alberts et al., 1999; Andersson, 2001; Ek et al., 1977; Rembetski et al., 2018). However, generation of spontaneous USM tone, demonstrated in excised urethral strips or rings from several species, is myogenic and occurs in the presence of tetrodotoxin and sympathetic receptor blockers (Brading, 1999). Despite the clinical importance of USM to urinary continence, cellular mechanisms of how USM tone is generated or how tone is increased or decreased by neural inputs or agonists are not well understood. Interventions to treat incontinence disorders stemming from urethral malfunction often rely on non-specific pharmacology with low patient compliance or invasive surgeries with limited long-term effectiveness (Kirschner-Hermanns et al., 2016; Staskin et al., 2011). Thus, understanding how USM tone is modulated could have valuable therapeutic potential to treat urethral obstruction (benign prostatic hyperplasia) or stress urinary incontinence (Delancey, 2010; DeLancey et al., 2008; Hashitani, Mitsui et al., 2024; Hokanson et al., 2017; Venema et al., 2023).

## Bladder anatomy and function

The bladder wall consists of both mucosal and muscular layers. The urothelium, the innermost layer of the bladder, plays a critical role in maintaining water tightness. While not entirely impermeable, it acts as a highly effective barrier, allowing for a very slow passage of water and electrolytes (Durnin et al., 2019). Between the basement membrane of the urothelium and the detrusor sits the lamina propria, a structure rich in vascular and lymphatic vessels, smooth muscle fascicles, elastic fibres, and containing numerous cell types (adipocytes, interstitial cells, fibroblasts) and nerve endings (Gabella, 2019b). Because of its rich and convoluted structure, the lamina propria fulfils multiples roles, including transport of oxygen and nutrients through blood vessels, production of collagen by fibroblasts and neuronal signal transduction to DSMCs (Andersson & McCloskey, 2014).

DSMC bundles interlace and intertwine, allowing the bladder body to contract in all directions simultaneously during micturition (Elbadawi, 1996). During bladder filling, there is a gradual increase in intravesical pressure, which triggers activation of afferent sensory nerves, believed to convey feelings of fullness to the central nervous system (CNS; de Groat & Yoshimura, 2009; Gabella, 2019a). Disrupted sensory feedback has been implicated in various bladder disorders (Araki et al., 2008), but specific mechanisms involved in the sensation of bladder fullness remain somewhat unclear. Afferent nerve endings are found in the detrusor and mucosal layers, but accumulating evidence suggests that the urothelium primarily dictates the activation of afferent nerves. In response to stretch, mechanoreceptors present on urothelial cells mediate the release of chemical stimuli (i.e. ATP) which diffuse and activate the underlying afferent nerves, contributing to a sensation of fullness in the bladder (Li et al., 2023a). However, cells within the detrusor layer also express mechanoreceptors (TRPV4; Vanneste et al., 2021) and recent evidence has shown that an emerging family of mechanoreceptors are expressed on both DSMCs (Piezo1) and afferent nerves (Piezo2); however, their precise functional role is still yet to be fully resolved (Li et al., 2023b). Therefore, it is still uncertain whether the detrusor layer plays a crucial role in perceiving fullness or sensing pressure changes during filling, despite expressing mechanoreceptors.

## Spontaneous activity and tone in urethral smooth muscle

In *ex vivo* preparations from pig and rabbit, USM strips develop spontaneous myogenic tone upon which rapid, low amplitude phasic contractions may be superimposed (Bradley et al., 2006; Bradley et al., 2010; Bridgewater et al., 1993; Rembetski et al., 2020). In rabbit and guinea pig USM, sharp electrode impalements demonstrated regular oscillations in resting membrane potential (Fig. 1), which were ablated by inhibitors of sarcoplasmic reticulum (SR) $Ca^{2+}$ release and $Ca^{2+}$-activated $Cl^-$ channels (Hashitani & Edwards, 1999; Hashitani et al., 1996). This activity was similar to electrical 'slow waves' in gastrointestinal (GI) tissues, which pace SMCs to generate motility (Chow & Huizinga, 1987; El-Sharkawy, 1983; Smith et al., 1987).

At the cellular level, USMCs isolated from proximal sheep urethra exhibited spontaneous transient inward currents (STICs) in voltage clamp; 10% of cells fired STICs when held at −60 mV (Sergeant et al., 2001b). Rhythmic intracellular $Ca^{2+}$ oscillations or propagating intracellular waves have been recorded from *in situ* rabbit USMC preparations loaded with Fluo 4-AM (Hashitani & Suzuki, 2007) and with mice expressing SMC-specific GCaMP3 (Drumm et al., 2018). Occurrence of spontaneous rhythmic $Ca^{2+}$ and electrical activity in USMCs raises two questions. Firstly, if USMCs are spontaneously rhythmic, how is the urethra a tonic organ? Secondly, is this an intrinsic property of USMCs or does it require input from another (non-SMC) cell type? A potential answer to the first question stems from observations that USMC bundles (and cells within bundles) are not well coupled electrically (Sancho et al., 2011), and exhibit asynchronous activity across tissues. When imaged *in situ*, USMCs are spontaneously active, firing regular $Ca^{2+}$ oscillations or propagating intra-

cellular Ca$^{2+}$ waves (Drumm et al., 2018; Hashitani & Suzuki, 2007). These signals are linked to small contractions of individual USMCs or bundles.

Spatial–temporal map analysis of USMCs visualized in mice expressing GCaMP3 demonstrated that Ca$^{2+}$ waves do not propagate intercellularly, even when USMCs are located adjacent to each other, but rather USMC activity occurs asynchronously across the bundle (Drumm et al., 2018). Due to the out-of-phase, asynchronous firing of multiple USMCs, this led to multiple small rhythmic contractions across the bundle over the same period, which averaged as uniform tone (Fig. 2). Application of agonists such as phenylephrine or ATP (which contract USM in a uniform manner under basal conditions; Bradley et al., 2010; Rembetski et al., 2018) did not increase intercellular coordination. Instead, agonists increased Ca$^{2+}$ wave firing frequency in individual cells. Electrical field stimulation (EFS) of USM leads to uniform contractions (Fedigan et al., 2017; Gupta

et al., 2024; Rembetski et al., 2020). However, Ca$^{2+}$ imaging experiments on rabbit USM loaded with Fluo 4-AM, or mice expressing GCaMP exclusively in USM, demonstrated that EFS generated this uniform response by increasing asynchronous USMC Ca$^{2+}$ oscillation frequency across muscle bundles, without increasing intercellular coordination (Gupta et al., 2024; Thornbury et al., 2011). These asynchronous firing patterns are observed not only in cells from the main USM walls but also in USMCs located in the lamina propria of pig urethra (Mitsui et al., 2021).

## Spontaneously active interstitial cells in urethra

The second question, as to whether USM spontaneous activity requires input from a non-SMC cell-type, first became relevant when rhythmic depolarization events were recorded from circular USM of rabbit and guinea-pig (Hashitani & Edwards, 1999; Hashitani et al., 1996). These

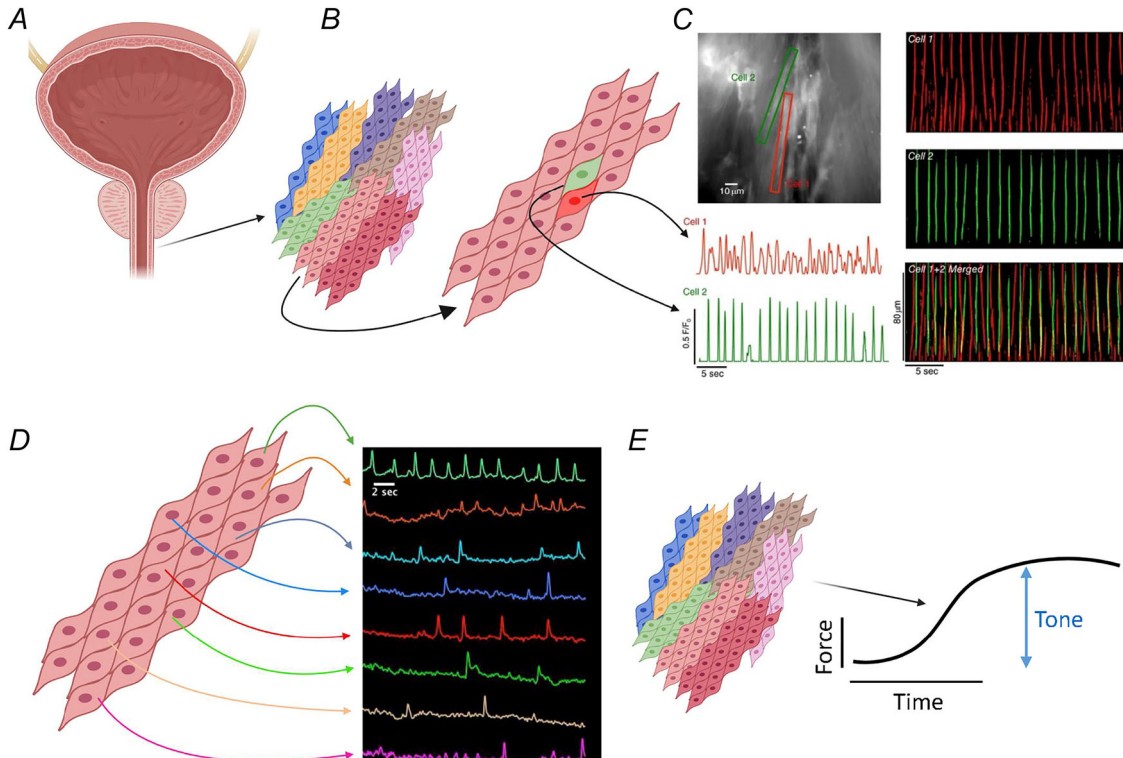

**Figure 2. Generation of urethral tone from asynchronous rhythmic activity**
*A*, the walls of USM that comprise the urethral sphincter consist of USM bundles that are arranged in a mesh like fashion along a rough longitudinal and circular orientation. *B*, USM bundles (shown in different colours) contain numerous USM cells (USMCs), whose activity is poorly coupled. *C*, even within a single bundle, two cells that are physical adjacent to each other (red and green cells) display spontaneous rhythmic activity in the form of intracellular propagating Ca$^{2+}$ waves that underlie contractions. Rises in Ca$^{2+}$ for two cells in the same bundle are shown in the bottom left of the panel, coloured red and green. When the Ca$^{2+}$ wave activity in both cells was plotted as a spatiotemporal map and superimposed upon each, there was no evidence of coordination of activity, despite the proximity of the cells to each other (Drumm et al., 2018). *D*, across an entire bundle, multiple USMCs can therefore display spontaneous Ca$^{2+}$ wave activity and resulting contractions that while rhythmic to varying degrees are asynchronous with each other (Drumm et al., 2018). *E*, across multiple bundles, these asynchronous activities summate and average as relatively uniform, tonic contractions.

authors noted these events resembled 'slow waves' in the GI tract. Slow waves are initiated and propagated by specialized pacemakers, interstitial cells of Cajal (ICC) (Sanders et al., 2026b), which express the tyrosine kinase receptor Kit as a differential marker from SMCs (Sanders et al., 2014). Pacemaker ICC are arranged in networks at the myenteric plexus (ICC-MY) between longitudinal and circular SMCs, to which ICC are electrically connected via gap junctions. The pacemaker conductance in ICC is a $Ca^{2+}$-activated-$Cl^-$ channel (Ano1) (Gomez-Pinilla et al., 2009; Hwang et al., 2009), and ICC-MY activate Ano1 and generate slow waves via intracellular $Ca^{2+}$ release, sustained by $Ca^{2+}$ influx (Baker, Hwang et al., 2021; Baker, Leigh et al., 2021; Drumm et al., 2017; Sanders et al., 2022; Zhu et al., 2015). $Ca^{2+}$-activated $Cl^-$ channels expressed in other interstitial cells, which label against antibodies for platelet-derived growth factor receptor $\alpha$, are also essential for pacemaking regular contractions of the smooth muscle walls of the renal pelvis (Grainger, 2026; Grainger et al., 2020; Grainger et al., 2022) and male reproductive tracts (Hashitani, Takeya et al., 2026; Kudo et al., 2024).

Experiments on rabbit demonstrated that two cell types were present in urethral tissues (Sergeant et al., 2000). USMCs formed the majority (85–90%) of isolated cell suspensions, while non-contractile interstitial cells (10%) stained positive for vimentin and were similar in morphology to ICC isolated from the GI tract (Langton et al., 1989). In later studies, these urethral interstitial cells were ascribed various names (interstitial cells, ICC, Cajal-like interstitial cells), but for this review, we will use the term ICC-like cell (ICC-LC) to include all of these terms. Rabbit ICC-LC were abundant in a $Ca^{2+}$-activated $Cl^-$ current and fired spontaneous transient depolarizations (STDs) in current clamp and STICs (mediated by $Cl^-$) in voltage clamp (Sergeant et al., 2000). STDs were similar in waveform, frequency and pharmacological sensitivity to slow waves observed in USM preparations (Hashitani & Edwards, 1999; Hashitani et al., 1996).

Subsequent studies further characterized mechanisms of rabbit urethral ICC-LC spontaneous activity, finding regular spontaneous $Ca^{2+}$ waves that were mediated by $Ca^{2+}$ release via ryanodine receptors, amplified by inositol 1,4,5-trisphosphate (IP₃) receptors and $Ca^{2+}$ influx via reverse mode $Na^+/Ca^{2+}$ exchange (Bradley et al., 2006; Drumm, Koh et al., 2014, Drumm, Sergeant et al., 2014; Drumm et al., 2015; Johnston et al., 2005). This regular $Ca^{2+}$ activity in rabbit urethral ICC-LC was coupled to opening of $Ca^{2+}$-activated $Cl^-$ channels (Ano1; Fedigan et al., 2017), which would result in $Cl^-$ efflux (resulting in STICs and STDs). Together, this suggested spontaneous activity in ICC-LC could increase the excitability of electrically coupled USM, leading to

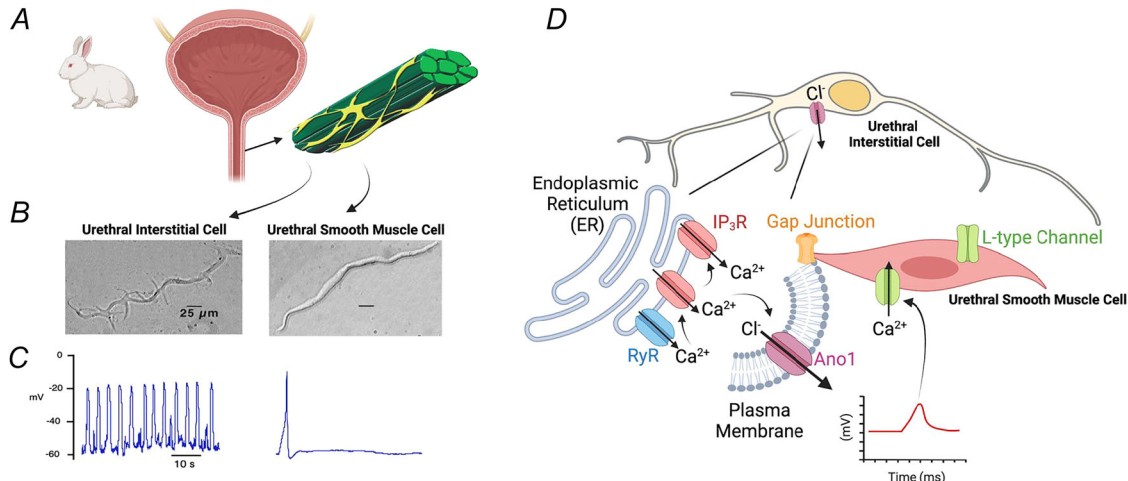

**Figure 3. Interstitial cells as pacemakers in the rabbit urethra**

*A*, enzymatic dispersal of USM tissues from the rabbit urethra reveal two distinct cell types. *B*, contractile urethral smooth muscle cells (USMCs) make up the majority of isolated cells (80–85%) and non-contractile interstitial cells making up the remainder (15–20%) (Sergeant et al., 2000). *C*, while rabbit USMCs are electrically quiescent, they can elicit action potentials in response to current injection, while rabbit urethral interstitial cells are spontaneously active, with regular depolarizations in membrane potential similar to electrical slow waves observed in whole tissues from rabbit and guinea-pig (Sergeant et al., 2000). *D*, rabbit urethral interstitial cells may serve as a pacemaker to increase the excitability of USMCs within USM bundles. Propagating $Ca^{2+}$ waves, generated by endoplasmic reticulum $Ca^{2+}$ release via ryanodine receptors (RyR) and IP₃ receptors (IP₃Rs) in interstitial cells activate a $Cl^-$ conductance encoded by Ano1, which depolarizes interstitial cells, and this is conducted to electrically coupled USMCs via gap junctions. This increases the open probability of voltage-dependent L-type $Ca^{2+}$ channels, resulting in $Ca^{2+}$ influx and contraction of USMCs.

USMC voltage-dependent $Ca^{2+}$ channel opening and contraction, similar to pacemaker ICC in the gut (Fig. 3).

Unequivocal identification of urethral ICC-LC has been challenging within intact tissues, as the marker used in initial studies, vimentin (Sergeant et al., 2000), labels multiple cells in urinary tract tissues (Koh et al., 2012). Attempts to label urethral ICC-LC with the gut ICC marker Kit proved difficult, as antibody labelling against this protein was initially elusive in the rabbit (Sergeant et al., 2000), and in rat and sheep urethra only weak Kit staining of vimentin$^+$ cells could be achieved (García-Pascual et al., 2008). However, later studies succeeded in labelling Kit$^+$ ICC-LC in rabbit urethra distinct from SMCs (Hashitani & Suzuki, 2007; Lyons et al., 2007; McHale et al., 2006). Mice expressing the genetically encoded $Ca^{2+}$ indicator GCaMP expressed exclusively in Kit$^+$ cells with Cre recombinase driven from Kit (c-Kit$^{+/Cre-ERT2}$) has allowed investigators to distinguish ICC-LC from SMCs in tissues where Kit antibody labelling of ICC-LC has proven challenging, such as the mouse urethra (Gupta et al., 2024) and renal pelvis (Grainger et al., 2020). Labelling mouse Kit-GCaMP tissues with antibodies against green fluorescent protein (GFP), enables Kit$^+$ ICC-LC expressing GCaMP to be visualized in immunohistochemical imaging, owing to antibody labelling of the GFP molecule, which is a constitute component of GCaMP. These approaches have shown that unlike pacemaker ICC-MY in the GI tract, which are arranged into networks, urethral ICC-LC are not interconnected (Gupta et al., 2024), and instead resemble distribution of intramuscular ICC (ICC-IM) which run along and between GI muscle bundles (Sanders et al., 2014).

Generation of tone in rabbit and pig USMCs is thought to rely on bringing USM resting membrane potential into an activation window for L-type $Ca^{2+}$ channels, which open to cause $Ca^{2+}$ influx and contraction (Brading, 2006). Rabbit urethral ICC-LC, through Cl$^-$ channel activation, might depolarize USMC bundles in a rhythmic manner (generation of local slow waves); this would increase USMC excitability into the window current activation range, opening L-type $Ca^{2+}$ channels, resulting in a relatively small number of USMC contractions in that bundle. Experiments with Kit-GCaMP models show that while multiple ICC-LC within a single urethral tissue exhibit dynamic spontaneous activity, with each cell displaying varying degrees of intrinsic rhythmicity in generating regular $Ca^{2+}$ transients, this occurs asynchronously across multiple cells and with no intercellular entrainment (Gupta et al., 2024). This observation, along with the known poor electrical coupling amongst USM bundles (Sancho et al., 2011), suggests that if spontaneous activity in ICC-LC contributes to overall tone, ICC-LC do so at a relatively local level, influencing activity of USMCs within the bundle to which it is connected. This tissue-wide, asynchronous USMC activity could manifest as sustained tone at the tissue level. This might explain why simultaneous $Ca^{2+}$ transient recordings from rabbit urethral ICC-LC and USMCs *in situ* show little correlation, as ICC-LC may not be 'pacing' USMCs in a traditional 1 to 1 sense, but rather boosting overall USMC excitability within the bundle (Hashitani & Suzuki, 2007). This is similar to models proposed for how phasic activity in ICC-IM leads to generation of tone in lower oesophageal sphincters (Drumm, Hannigan et al., 2022) and internal anal sphincters (Hannigan et al., 2020).

## Species differences in USM spontaneous activity

The question of USM tissues generating spontaneous activity and resultant tone or neural responses is complicated, owing to species differences between cellular behaviours and expression of ionic conductances (Brading, 2006). Spontaneous activity in ICC-LC has been most extensively studied in the rabbit as described above. However, unlike the rabbit, sheep USMCs possess Cl$^-$ currents and fire Cl$^-$-mediated STICs under voltage clamp (Sergeant et al., 2001b). In rabbit, vimentin$^+$ ICC-LC express the Ano1 channel almost exclusively, but in mouse, rat and sheep, Ano1 can be detected in USMCs (Sancho et al., 2012). Limited data from interstitial cells isolated from human urethra show these cells and not USMCs express Cl$^-$ currents (Drumm et al., 2021). This suggests that some models of urethral spontaneous activity from rabbits may be applicable to humans, but this must be stated cautiously and a rigorous examination of this possibility in humans is warranted.

The importance of the Ano1 channel itself in regulating urethral rhythmicity and tone also differs among species. While Cl$^-$ channel inhibitors reduced tone or EFS evoked contractions of USM from rabbit, mouse, rat and sheep (Fedigan et al., 2017; Sancho et al., 2012), more specific Ano1 channel inhibitors such as Ani9 do not affect tone or EFS responses from pig urethra, which were sensitive to the L-type $Ca^{2+}$ channel blocker nifedipine (Rembetski et al., 2020). It is possible these discrepancies represent differences in chosen pharmacology. Previous experiments from rabbit, mouse, rat and sheep utilized drugs such as niflumic acid, 4,4′-diisothiocyano-2,2′-stilbenedisulfonic acid, CaCCinhA01 and T16AinhA01, which are now known to exert myriad off-target effects including modulation of $Ca^{2+}$ influx and SR $Ca^{2+}$ release mechanisms (important for USMC activity; Fig. 4), whereas the potent and selective Ano1 inhibitor Ani9 (Seo et al., 2016) does not display such non-specific effects (Danahay et al., 2023; Dwivedi et al., 2023; Genovese et al., 2023; Lim et al., 2022). Future experiments on animal models should re-examine more carefully pharmacological verification

as well as cell-specific Ano1 ablation to fully resolve this question.

As described in previous sections, USM tone is built on summation of asynchronous cellular activity across USMCs in multiple bundles (Fig. 2). In this model, USMC contraction relies on intracellular $Ca^{2+}$ signalling, presumed to be through $Ca^{2+}$ influx via L-type $Ca^{2+}$ channels. Voltage-gated $Ca^{2+}$ channels such as L-type and T-type $Ca^{2+}$ channels are expressed in USMCs from numerous species, including humans (Hollywood et al., 2003), and L-type $Ca^{2+}$ channel inhibitors, such as nifedipine, significantly inhibit tone of rabbit (Brading, 2006) and pig urethra (Greenland et al., 1996; Rembetski

et al., 2020). However, in male mice, spontaneous USMC $Ca^{2+}$ waves underlying asynchronous contractions of USM bundles are insensitive to L-type channel blockers, and instead are generated by SR-mediated $Ca^{2+}$ release sustained by store-operated $Ca^{2+}$ entry (SOCE) via STIM–Orai interactions (Fig. 4; Drumm et al., 2018). Relative contributions of L-type channels or STIM–Orai $Ca^{2+}$ influx pathways to spontaneous rhythmicity in female USMCs or to EFS or agonist evoked responses in either sex has not been investigated and warrants careful study, as male and female USM display differences in responses to identical neural/agonist stimuli (Alexandre et al., 2017; Drumm, Griffin et al., 2022), which may reflect

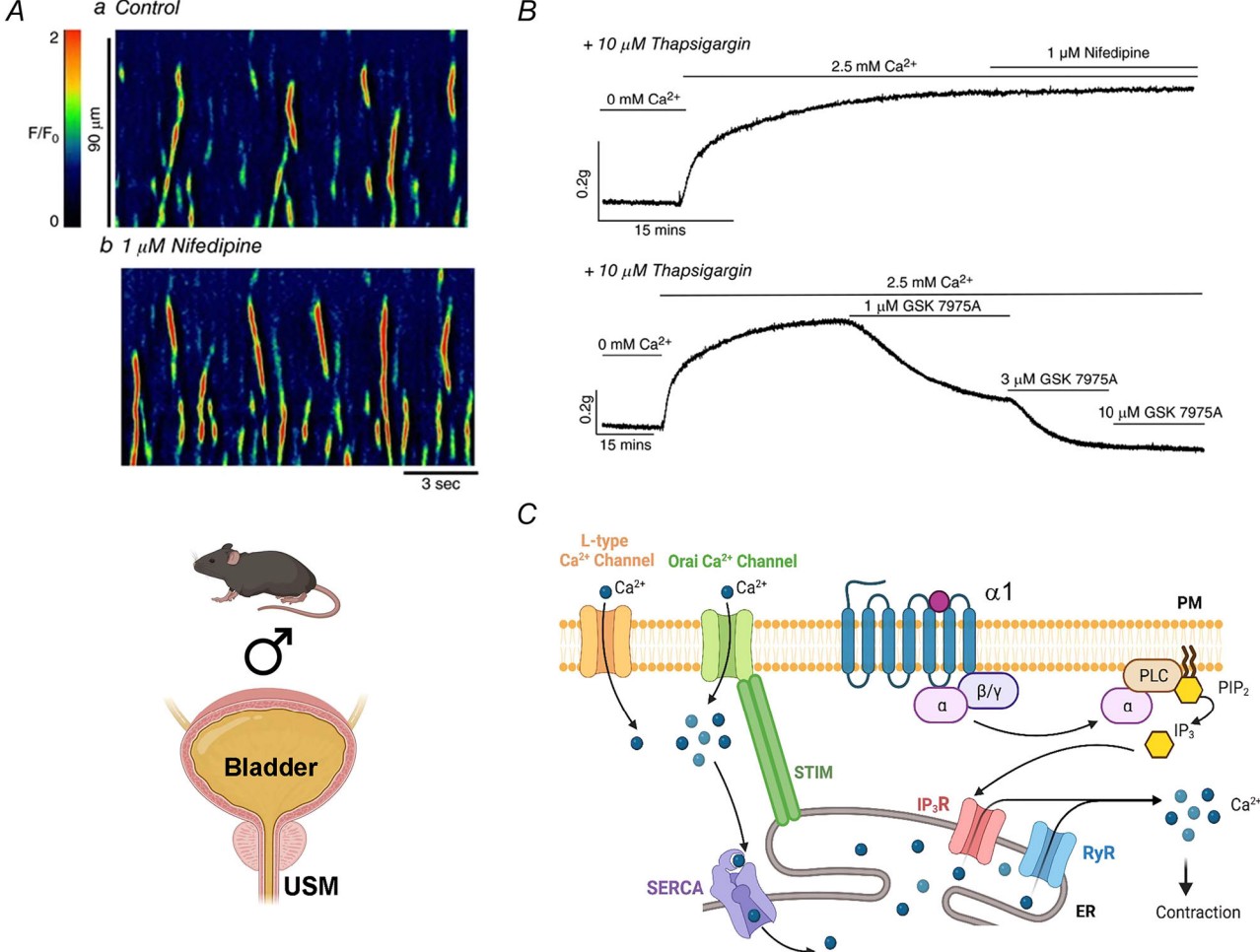

**Figure 4. Spontaneous $Ca^{2+}$ activity and contractions of mouse urethra do not require L-type $Ca^{2+}$ channels but depend on $Ca^{2+}$ influx via store-operated $Ca^{2+}$ entry (SOCE)**
*A*, spatio-temporal maps of spontaneous $Ca^{2+}$ waves recorded from USMCs *in situ* from male mice expressing GCaMP3 in SMCs. The L-type $Ca^{2+}$ channel inhibitor nifedipine had no effect on spontaneous activity (Drumm et al., 2018). *B*, in male mouse USM tissues, reintroduction of $Ca^{2+}$ following store depletion with $Ca^{2+}$ free solution and thapsigargin leads to generation of sustained tonic contractions that are resistant to nifedipine but dose-dependently reduced by the store-operated $Ca^{2+}$ entry antagonist GSK 7975A (inhibits SOCE Orai channels) (Drumm et al., 2018). *C*, in male mice, contraction of USMCs does not rely on $Ca^{2+}$ influx via L-type channels but rather interactions between plasma membrane located Orai channels and ER located STIM proteins that initiate SOCE. SOCE maintains regular $IP_3$ mediated ER $Ca^{2+}$ release (increased by activation of $\alpha1$ receptors) which activates contractile machinery in USMCs.

underlying differences in $Ca^{2+}$ handling mechanisms, ionic conductances or distribution of 'interstitial' cell types.

In mouse, urethral tone and agonist responses do not rely on $Ca^{2+}$ influx via voltage-dependent $Ca^{2+}$ channels (Drumm et al., 2018), which is different from that observed in the rabbit or pig urethra (Brading, 1999; Drumm, Sergeant et al., 2014; Greenland et al., 1996). Furthermore, while ICC-LC and USMCs express gap junctions, specifically connexin 43 and 37 in rat and sheep urethra (Sancho et al., 2011), this coupling is relatively poor. This raises the question of how spontaneous activity in ICC-LC might influence the activity of USMCs if in certain species, such as mouse, voltage-dependent $Ca^{2+}$ influx is not required for USMC activity.

In mouse, if ICC-LC influence USMCs it is possibly through a mechanism independent of membrane potential. Such mechanisms might include paracrine release of a $Ca^{2+}$-dependent messenger molecule from ICC-LC to USMCs. For example, ATP increases basal urethral tone and USMCs in rabbit and mouse (Bradley et al., 2010; Drumm et al., 2018). ATP can be released from one cell via vesicles or travelling through pannexin or connexin channels and act on another cell (Leybaert & Sanderson, 2012). Such mechanisms are proposed to account for non-electrical intercellular communication between enteric glia and neurons in colon (Grubisic & Gulbransen, 2017a, b). Urethral ICC-LC respond to exogenous ATP via activation of $P_2Y_1$ receptors (Bradley et al., 2010), but it is unknown if ICC-LC are capable of secreting ATP themselves. If they are, it is possible that paracrine ATP could act on neighbouring USMCs to increase $Ca^{2+}$ release, which underlies USMC contraction.

Alternatively, $Ca^{2+}$ itself or $IP_3$ could diffuse through gap junctions and act as an intercellular second messenger (Boitano et al., 1992). $Ca^{2+}$ activity is abundant in ICC-LC *in situ* (Gupta et al., 2024) and such activity likely represents dynamic fluctuations in intracellular $IP_3$ (Johnston et al., 2005; Sergeant et al., 2001a). Diffusion of either $Ca^{2+}$ or $IP_3$ from ICC-LC might raise cytosolic USMC $Ca^{2+}$ or $IP_3$ concentrations. This would create a more excitable cytosolic pool of $Ca^{2+}$ signalling (Dupont & Goldbeter, 1994; Sneyd et al., 1995; Sneyd et al., 2017; Wacquier et al., 2019), resulting in increased sensitivity of $IP_3Rs$ on the SR surface of USMCs, increasing the likelihood of SR $Ca^{2+}$ release, culminating in more regular or forceful USMC contractions. Intercellular diffusion of ATP, $Ca^{2+}$ or $IP_3$ is now proposed to explain how certain pancreatic acinar cells 'pace' connected cells in the absence of traditional electrical pacemaking involving entrainment of voltage-dependent $Ca^{2+}$ channels (Takano & Yule, 2023), as well as signalling switching between ATP diffusion and re-generative $IP_3$ production to drive intercellular communication between vascular endothelium (Buckley et al., 2024). While evidence for such mechanisms in the urethra is lacking and requires rigorous experimentation to elucidate, these possibilities provide alternatives of how ICC-LC may increase USMC $Ca^{2+}$ activity and contraction independent of membrane potential.

In order to conclusively determine whether ICC-LC influence USMCs, there must be an evaluation of how selective stimulation or deletion of ICC-LC affects urethral function. Optogenetic stimulation of mouse tissues expressing light-sensitive channel rhodopsin (ChR2) exclusively in $Kit^+$ cells with Kit driven inducible Cre recombinase Kit-(iCre) mice may provide such a means of direct stimulation as shown for ICC in colon (Zhao & Tong, 2023) and ICC-LC in lymphatic tissues (Zawieja et al., 2023). However, such stimulation still relates to changes in membrane potential and might not be optimal in the mouse urethra. Alternatively, ICC-LC could be optogenetically stimulated using cell specific expression of photoactivated plasma membrane G-protein-coupled receptors linked to increased phospholipase C and $IP_3$ production, or photoactivated STIM/Orai proteins to increase SOCE (Ma et al., 2017). This could provide insight into how enhancement of ICC-LC activity might affect urethral contractions independently of membrane potential.

## Spontaneous 'rhythmical' activity in the bladder

The contractile force required for micturition consists of coordinated whole bladder contractions brought about by parasympathetic innervation of DSMCs (de Groat et al., 2015; Hill, 2015; Rembetski et al., 2018). During filling, DSMCs remain relatively relaxed at the whole organ level. However, as the detrusor wall expands, rhythmic and transient low-amplitude contractions are present, termed 'non-voiding' contractions (NVCs or transient contractions; Heppner et al., 2016). NVCs coincide with bladder afferent nerve activity. Consequently, as the detrusor wall expands and stretches, NVC activity increases, relaying a sensation of bladder fullness to the CNS (Heppner et al., 2016). While neurogenic input can alter the amplitude and frequency of NVCs, it appears this activity itself originates from non-neurogenic sources.

*Ex vivo* contractility experiments of isolated mucosa-free bladder strips display 'spontaneous' rhythmic low-amplitude contractions. It is important to note that 'NVC' typically refers to the context of an intact bladder preparation where phasic changes in pressure are detected (Andersson et al., 2011), whereas 'spontaneous contractions' refer to those present in excised isolated bladder strips where the influence of intravesical pressure is absent (Kullmann et al., 2014). Studies have found that both NVCs and spontaneous contractions are insensitive to tetrodotoxin, a phenomenon common

to rats (Kanai et al., 2007; Kullmann et al., 2014) and guinea pig (Hammad et al., 2014; Imai et al., 2001). These data led to the development of the 'myogenic theory' proposing the origin of spontaneous activity in bladder is inherent to DSMCs. Under pathophysiological conditions, spontaneous rhythmical contractile activity of isolated detrusor strips is greatly increased in patients with overactive bladder (OAB) (Brading, 1997; Hristov et al., 2013; Oger et al., 2011).

Spontaneous contractions of the bladder wall are directly preceded by action potentials and rises in intracellular DSMC $Ca^{2+}$ (Hashitani et al., 2001; Hashitani, Brading et al., 2004). This was first shown by Hashitani and colleagues, who simultaneously recorded spontaneous contractions, electrical activity, and $Ca^{2+}$ dynamics in isolated bladder sheet preparations from guinea-pig (Fig. 5*A*). Enzymatic dispersion of individual DSMCs reveals DSMCs generate regular changes in membrane potential and spontaneous action potentials independent of neural input (Anderson et al., 2013). These data suggest the origin of spontaneous contractile activity lies within DSMCs and perhaps not in a secondary 'pacemaker' cell.

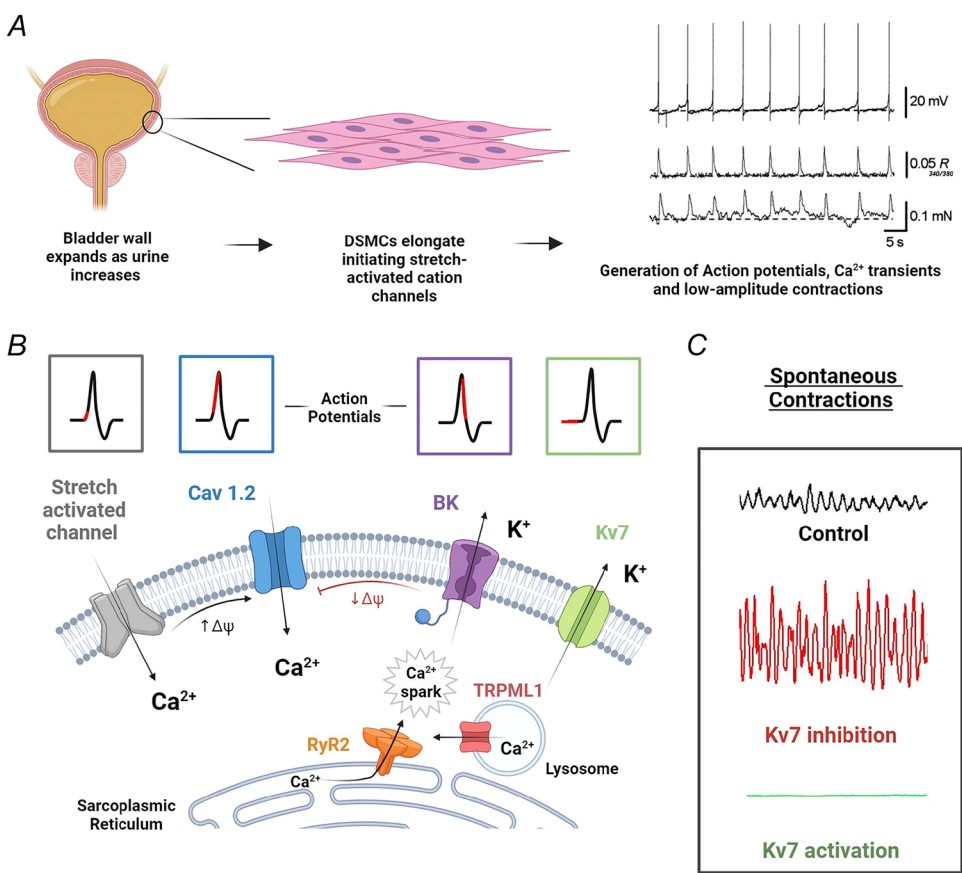

**Figure 5. Spontaneous myogenic activity of bladder smooth muscle and their primary ionic conductances**

*A*, during the filling phase, bladder smooth muscle (detrusor) displays 'spontaneous' rhythmic low-amplitude contractions in response to stretch, enabling the baldder to contract spontaneously irrespective of neural stimulation. Spontaneous contractions of the bladder wall are directly correlated and preceded by action potentials and rises in intracellular DSMC $Ca^{2+}$ (Hashitani, Brading et al., 2004). *B*, generation of spontaneous rhythmicity in the bladder is initiated, at least in part, by stretch-activated cation channels present on the membrane of DSMCs. Therefore stretch depolarizes membrane potential and opens L-type $Ca^{2+}$ channels (Cav1.2), which are pivotal for the upstroke component and generation of an action potential. Several hyperpolarizing influences, exist to relax DSMCs such as the large conductance $Ca^{2+}$-activated $K^+$ (BK) channels and voltage gated $K^+$ (KV7) channels. BK channels are activated by intracellular $Ca^{2+}$ sparks and are greatly involved in the repolarization phase of action potentials, whereas KV7 activation hyperpolarizes membrane potential opposing Cav1.2 open probability. *C*, isometric tension traces showing the profound effects of both activation and inhibition of KV7 channels on spontaneous rhythmic contractions. (Adapted from Fong et al. 2023).

## Ionic conductances contributing to DSMC spontaneous activity

Generation of spontaneous rhythmicity in the bladder is initiated, at least in part, by stretch. DSMCs express stretch-activated, non-selective cation channels that are activated upon elongation of the cells during bladder filling (Wellner & Isenberg, 1993a, b). Stretch activation of these channels leads to depolarization and enhanced open probability of L-type $Ca^{2+}$ channels ($Ca_v1.2$) and generation of an action potential (Wellner & Isenberg, 1993b). $Ca_v1.2$-mediated action potentials propagate intercellularly across DSMCs via gap junctions (Connexin) (Hashitani, Brading et al., 2004). Connexin 45 is localised to the detrusor layer, whereas Connexin 43 is more evident in the suburothelial mucosa (Ikeda et al., 2007; Sui et al., 2003). Functionally, $Ca_v1.2$ plays a major role in regulating spontaneous activity of bladder. For example, application of the Cav 1.2 channel antagonist nifedipine abolished the upstroke of action potentials in isolated DSMCs (Hashitani et al., 2000; Hashitani et al., 2001; Kajioka et al., 2002). Indeed, this effect translates to contractility studies, where nifedipine significantly diminishes spontaneous phasic contractions of porcine and guinea-pig bladder (Buckner et al., 2002; Herrera et al., 2000), and genetic ablation of $Ca_v1.2$ specifically in smooth muscle results in complete abolishment of any spontaneous activity (Wegener et al., 2004). Therefore, it appears propagation of electrical activity (i.e. action potentials) and $Ca^{2+}$ transients underlying bladder spontaneous activity is mediated by L-type $Ca^{2+}$ channels expressed in DSMCs (Hashitani & Brading, 2003a, b).

For normal frequencies of bladder rhythmicity to be maintained, opening of DSMC $Ca_v1.2$ channels must be impacted by hyperpolarizing influences, thus allowing sufficient intervals for recovery/relaxation of DSMCs between phasic events. Several $K^+$ channels expressed on DSMCs are implicated in modulating the duration and shape of underlying action potentials and DSMC spontaneous contractions (Afeli et al., 2013; Hristov et al., 2011; Malysz et al., 2013; Petkov, 2011). The $Ca^{2+}$-activated large conductance $K^+$ channel (BK) is one the most important regulators of DSMC excitability. Pharmacological inhibition of BK channels has profound effects on spontaneous contractions and action potentials of DSMCs. Recordings of electrical activity show blockade of BK channels prolongs action potential duration while increasing amplitude, suggesting BK channels are involved in the repolarization phase (Hashitani & Brading, 2003b). In human DSMCs, and those of various other species, pharmacological inhibition of BK channels significantly enhances amplitude, duration and force of spontaneous phasic contractions (Herrera et al., 2000; Hristov et al., 2011; Imai et al., 2001; Ohi et al., 2001). In contrast, application of BK channel agonists to bladder tissue inhibits spontaneous contractions (Large et al., 2015; Layne et al., 2010).

Several studies have identified 'Ca²⁺ sparks' as the underlying signal activating BK channels during the generation of bladder spontaneous contractions (Griffin et al., 2020; Heppner et al., 2003; Herrera et al., 2001; Ohi et al., 2001). $Ca^{2+}$ sparks are rapid, large-amplitude, highly localized, spontaneous $Ca^{2+}$ signals originating from clusters of ryanodine type 2 receptors (RyR2s) on the SR of SMCs (Jaggar et al., 2000). In DSMCs, RyR2s and $Ca^{2+}$ spark activity are functionally coupled to nearby BK channels, giving rise to large-amplitude outward currents known as spontaneous transient outward currents (STOCs), leading to hyperpolarization of membrane potential and reduced muscle excitability (Herrera et al., 2001). Inhibition of RyR2 and $Ca^{2+}$ sparks leads to abolishment of BK-channel–STOC activity, enhancing spontaneous contractions and underlying action potentials (Hashitani & Brading, 2003b; Herrera et al., 2001).

Both mice lacking the BK channel pore-forming $\alpha$-subunit (Meredith et al., 2004; Thorneloe et al., 2005) and heterozygous RyR2-knockout mice (Hotta et al., 2007) exhibit absent or decreased STOCs, enhanced amplitude of spontaneous contractions, and increased urination frequency. In patients with detrusor over-activity, an almost 16-fold reduction in BK channel $\alpha$-subunit mRNA expression has been detected (Hristov et al., 2013). Therefore, current evidence suggests a critical role for the BK channel/$Ca^{2+}$ spark pathway in regulating DSMC spontaneous activity, and its potential involvement in the pathophysiology of bladder overactivity. Recently, lysosomal TRPML1 channels have been implicated in the initiation of intracellular $Ca^{2+}$ sparks in DSMCs. Muscle strips taken from TRPML1 KO mice showed greatly enhanced spontaneous contractions, while iso-lated DSMCs from these mice lacked $Ca^{2+}$ sparks or BK-mediated STOCs. TRPML1 KO mice also displayed a remarkable increase in urination frequency, a phenotype indicative of OAB (Griffin et al., 2020).

An additional $Ca^{2+}$-activated $K^+$ conductance has been implicated in regulating bladder excitability. The small conductance $Ca^{2+}$-activated $K^+$ channel (SK) is involved in regulating the duration of the afterhyperpolarization phase of detrusor action potentials (Hashitani & Brading, 2003b). Using sharp electrode electrical recordings of detrusor, application of the SK channel blocker apamin converted single action potentials into bursts (Hashitani & Brading, 2003a, b; Hashitani, Brading et al., 2004). Iso-metric tension contractility experiments revealed apamin enhanced bladder spontaneous contractions (Afeli et al., 2012; Herrera et al., 2000; Herrera et al., 2005; Thorneloe et al., 2008). Upregulation of SK channel subtype 3 (SK3) expression in mice led to increased bladder capacity and reduced frequency of both spontaneous contractions

and NVCs. Conversely, the same study showed that suppressing expression of SK3 resulted in more frequent spontaneous contractions and NVCs, manifesting an OAB phenotype (Herrera et al., 2003). Although there is a large body of evidence supporting SK channels as regulators of bladder spontaneous contractions, interestingly at the single cell level, DSMCs possess little to no SK channel activity. Murine DSMCs show low SK current density (0.5 pA/pF) compared to other cell types in the bladder, with no current detectable at physiological potentials ($-40$ mV; Lee et al., 2013). Therefore, effects of SK channel inhibition on bladder excitability are ascribed to cell types other than DSMCs (see next section).

The voltage gated class of $K^+$ channels ($K_v$) are also critical regulators of spontaneous bladder contractions. The lack of selective antagonists renders study of individual $K_v$ subtypes difficult, with most studies utilizing non-selective agents such as tetra-ethylammonium (TEA), although $K_v$ subtype 7 ($K_v7$) has relatively selective pharmacological agents (Petkov, 2011). Depending on the species, various expression profiles of $K_v7$ subtypes ($K_v1$–$K_v7$) have been detected in the bladder (Malysz & Petkov, 2020). *In vitro* experiments on isolated bladder strips from mouse, rat, pig and human demonstrated that a non-selective $K_v7$ inhibitor, XE991, enhanced spontaneous contractions, whereas a $K_v7$ activator reduced activity (Rode et al., 2010; Svalo et al., 2013; Svalo et al., 2015).

At the single cell level, the $K_v7$ activator retigabine hyperpolarized membrane potential of isolated guinea-pig DSMCs and reduced spontaneous action potential frequency (Afeli et al., 2013; Provence et al., 2015). It should be noted that some have suggested that relaxation effects of retigabine on the bladder are due to off-target effects on voltage-dependent $Ca^{2+}$ channels and $K_v2$ channels on afferent nerves (Tykocki et al., 2019). Pharmacological activation of $K_v7$ reduced intracellular $Ca^{2+}$ concentrations in isolated DSMCs, and this effect was prevented once $Ca_v1.2$ channels were blocked with nifedipine (Provence et al., 2015). These findings suggest that $K_v7$ channels play a significant role in regulating spontaneous bladder contractions, most likely by acting as a hyperpolarizing force opposing open probability of $Ca_v1.2$ channels. The role of the other $K_v$ family subtypes needs to be studied in more detail before conclusions can be drawn about their contributions to spontaneous bladder contractions. Future development of $K_v$ family subtype specific pharmacology will greatly help in this effort. Overall, several molecular mechanisms outlined here provide DMSCs with the ability to generate and propagate spontaneous activity through the bladder muscular wall (Fig. 5).

## Evidence of interstitial cell-mediated pacemaking in the bladder

A significant body of evidence discussed in this review has attested that spontaneous rhythmic detrusor activity initiates in DSMCs, but a population of interstitial cell (IC)-like cells exist in the bladder and are implicated in regulating spontaneous activity (Fry & McCloskey, 2019; Koh et al., 2018). For almost 20 years, the nature and function of bladder IC-like cells has been controversial, and a consensus on their physiological importance is still debated. IC-like cells were first identified in bladder through immunostaining, when McCloskey & Gurney (2002) showed significant cellular c-Kit positivity in guinea pig bladder. These cells displayed stellate morphology, while running parallel with SMCs peripheral to muscle bundles, reminiscent of ICC-IM in the GI tract (Blair et al., 2012). Subsequent studies reported c-Kit positivity in bladder of other species such as rat, mouse and human (Kim et al., 2011; Shafik et al., 2004; Yu et al., 2012), and c-Kit$^+$ cells also co-stained for vimentin (McCloskey et al., 2009).

Bladder IC-like cells were initially speculated to have similar functions as ICC in the GI tract, as initial single cell studies identifying the ionic conductances and cellular activity present in c-Kit$^+$ bladder cells suggested they could serve as pacemakers. For example, enzymatic dispersal of branch-like c-Kit$^+$ cells from bladder revealed the cells displayed spontaneous $Ca^{2+}$ waves (McCloskey & Gurney, 2002), possessed robust L-type and T-type $Ca^{2+}$ currents (McCloskey, 2006), and could elicit $Ca^{2+}$ transients with cholinergic agonists that were predominantly dependent on store release via $IP_3Rs$ and RyRs. These findings suggested that c-Kit$^+$ IC-like cells could propagate depolarizing signals to DSMCs, initiating action potentials underlying spontaneous bladder activity. However, an elegant study by Hashitani, Yanai et al. (2004) suggested this was not the case. This study employed whole-tissue *in situ* imaging of $Ca^{2+}$ activity of both IC-like cells and DSMCs simultaneously, to measure synchronicity of $Ca^{2+}$ events between both cell types. They found fewer than 10% of IC-like cells displayed spontaneous $Ca^{2+}$ waves and this activity occurred at significantly lower frequencies than the $Ca^{2+}$ transients generated and propagated in DSMCs. Interestingly, activity generated by IC-like cells was insensitive to nifedipine, even though robust L-type currents were reported in isolated IC-like cells (McCloskey, 2006).

At the functional level, contractility and microelectrode studies have been performed on mutant mice that possess reduced tyrosine kinase function ($W/W^v$), an intervention that leads to lesions in ICC. Historically, $W/W^v$ mice have been utilized to display the critical function of ICC in the GI tract (Burns et al., 1996; Sanders et al.,

2010; Ward & Sanders, 2001; Ward & Sanders, 2006; Ward et al., 1994; Ward et al., 2004). Isolated bladders from *W/W^v* mice showed no significant change in resting membrane potential, spontaneous action potential activity or contractile properties when compared to wild-type mice (McCloskey et al., 2009). Ano1 serves as a signature functional ion channel mediating ICC-mediated slow wave activity of the GI tract (Gomez-Pinilla et al., 2009; Hwang et al., 2009; Singh et al., 2014; Zhu et al., 2009). In the bladder, non-selective Ano1 antagonists such as niflumic acid show inhibitory effects on juvenile rat spontaneous contractions (Bijos et al., 2014), however in adult rats niflumic acid enhances this activity, suggesting that a Cl$^-$ conductance normally underlying GI tract pacemaking is not fundamental for spontaneous rhythmic activity in bladders of matured rats (Lam et al., 2014). In the guinea-pig, a suppopulation of IC-like cells in the lamina propria exhibit a Cl$^-$ conductance without displaying an inward Ca$^{2+}$ current. These cells respond to ATP and have been proposed to regulate afferent nerve activity in the mucosa (Wu et al., 2004), but evidence for these cells acting as primary pacemakers has not been established yet. Therefore, there remains to be definitive functional evidence that IC-like cells serve as pacemakers initiating bladder spontaneous activity.

Further studies provided evidence that c-Kit immuno-labelling is an ineffective tool for detecting IC-like cells in the bladder since mast cells in this tissue also express c-Kit (Gevaert et al., 2017). A study by Koh et al. (2012) tested numerous commercially available c-Kit-targeting antibodies on mouse bladder tissues and detected little fluorescence, while identically processed GI tract tissues displayed typical populations of c-Kit$^+$ ICC.

Therefore, it appears that typical ICC identification markers such as c-Kit are unreliable for exploring pacemaker cells in the bladder. In the GI tract, a population of interstitial cells exist that are c-Kit negative

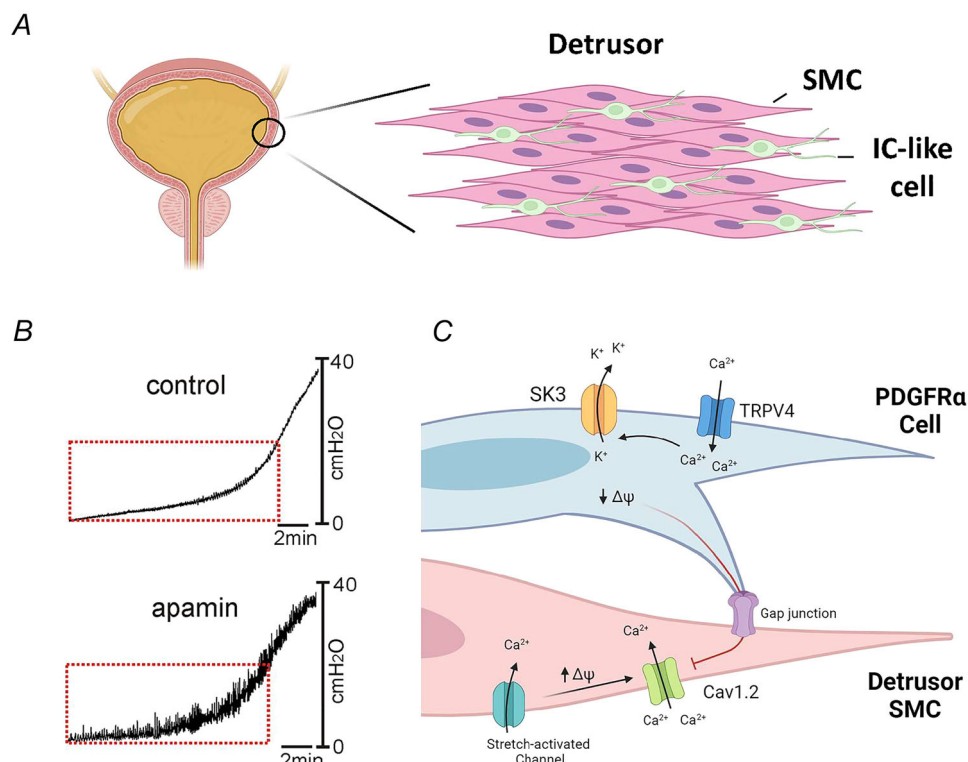

**Figure 6. Interstitial-like cells in the detrusor**
*A*, the detrusor layer of the bladder wall contains a population of IC-like cells with stellate, branched morphology that are distributed along the edges of the smooth muscle bundles. *B*, ex vivo bladder cytometric recordings showing typical bladder spontaneous rhythmic contractions (red dotted rectangle) present in the top trace, while the bottom trace shows contractions that are greatly enhanced upon addition of the SK channel blocker, apamin. This effect is thought to be brought about by inhibition of SK channels present on bladder IC-like PDGFR*α* cells. (From Lee et al., 2017.) *C*, TRPV4 channels present on detrusor PDGFR*α* cells activate in response to stretch, giving rise to elevations in Ca$^{2+}$ that stimulate nearby SK3 channels releasing K$^+$ that causes hyperpolarization of membrane potential ($\Delta\psi$) which is conducted to electrically coupled DSMCs. This hyperpolarizing signal restrains the open probability of L-type Ca$^{2+}$ channels present on DSMCs. Therefore, PDGFR*α* cells serve to negatively 'pace' bladder spontaneous rhythmic contractions by restraining DSMC membrane potential via the mechanism shown.

but electrically coupled to SMCs. These cells are identified by their immunopositivity to platelet derived growth factor receptor $\alpha$ (PDGFR$\alpha$) and function to counter-balance ICC-pacemaker-mediated activity by providing hyperpolarizing signals to GI tract SMCs (Sanders et al., 2023a). In mouse bladder, PDGFR$\alpha^+$ cells also co-label with vimentin, and this fact, along with the difficulty in reliable Kit staining for non-mast cell types in the bladder, suggests that previous studies that used solely vimentin staining to identify ICC-LC in the bladder may have actually been studying PDGFR$\alpha^+$ cells and those that solely relied on Kit staining were likely visualizing mast cells (Koh et al., 2012).

Functional studies revealed that like in the GI tract (Baker et al., 2013; Kurahashi et al., 2011; Kurahashi et al., 2014), bladder PDGFR$\alpha^+$ cells provide a hyper-polarizing influence to electrically coupled DSMCs (Koh et al., 2018; Lee et al., 2013, 2014). PDGFR$\alpha^+$ cells are present and distributed along the edges of detrusor muscle bundles and lie within the lamina propria and submucosa of the bladder wall of mouse, guinea pig and human tissues (Lee et al., 2013; Monaghan et al., 2012). Lee et al. (2013) first performed electrophysiological studies on enzymatically isolated bladder PDGFR$\alpha^+$ cells, identifiable from DSMCs by their expression of GFP fused to nuclei in PDGFR$\alpha^+$ cells. Isolated PDGFR$\alpha^+$ cells lacked any significant voltage-dependent inward currents but possessed STOCs that were insensitive to BK channel blockers but inhibited by the SK channel antagonist apamin. Interestingly, at physiological potentials ($-40$ mV) no resolvable SK currents were detected in isolated DSMCs, and quantitative transcriptional analysis of sorted PDGFR$\alpha^+$ cells showed enriched SK3 expression (Lee et al., 2013). This implies that effects of SK channel blockade on bladder spontaneous contractions (Afeli et al., 2012; Herrera et al., 2000; Herrera et al., 2005; Thorneloe et al., 2008) could be attributed to PDGFR$\alpha^+$ cells.

Further electrophysiological studies showed that PDGFR$\alpha^+$ cells hyperpolarized in response to TRPV4 channel agonists, which were absent when SK channels were blocked, indicative of the existence of a functional relationship between both channels (Lee et al., 2017). Indeed, *ex vivo* cytometry experiments demonstrated that apamin and the TRPV4 antagonist RN-1734 increased bladder spontaneous contractions. The spontaneous activity present in RN-1734 was similar to activity present in bladders of TRPV$^{-/-}$ mice (Lee et al., 2017). The data presented in these studies lead to the hypothesis that TRPV4 channels on PDGFR$\alpha^+$ cells activate in response to stretch, giving rise to elevations in $Ca^{2+}$ that stimulate nearby SK3 channels, causing hyperpolarization that is conducted to electrically coupled DSMCs (Fig. 6C). This hyperpolarizing signal restrains the open probability of L-type $Ca^{2+}$ channels on DMSCs and reduces the generation of action potentials and spontaneous rhythmic contractions. Therefore, evidence outlined in this section suggests PDGFR$\alpha^+$ cells in bladder lack the ability to act as primary pacemakers; however, they appear to negatively 'pace' spontaneous rhythmic activity by stabilizing DSMC membrane potential and excitability during bladder filling.

## Commonalities and differences between DSM and USM rhythmic behaviours

In conclusion, smooth muscle of the bladder and urethra exhibits rhythmic cellular and tissue level activity manifesting as asynchronous $Ca^{2+}$ transients across bundles that summate as tone in the urethra, and rhythmic action potentials, $Ca^{2+}$ rises and contractions in the detrusor. In both organs, rhythmicity occurs asynchronously across muscle bundles, with little evidence of coordinated activity at the cellular or tissue level (for the bladder, this changes at the onset of micturition when a coordinated, cholinergic mediated contraction is required for proper bladder emptying). In the urethra, certain species appear to have a non-USMC cell type that might serve as an excitatory influence to USMC, through activation of Ano1 channels that increases open probability of voltage-dependent $Ca^{2+}$ channels in USMC, resulting in contraction. However, in mice at least, voltage-dependent $Ca^{2+}$ channels do not appear to play a role in generating basal urethral tone or response to neurotransmitters. The importance of Ano1 channels in urethral function also appears to vary across species, and thus a need for pacemakers or the function of interstitial cells present in these animals is not immediately obvious. In the bladder, the importance of voltage-dependent $Ca^{2+}$ channels to spontaneous activity is consistent across numerous species, including humans. While interstitial cells do exist in the bladder, the pacemaker role of c-Kit$^+$ IC-like cells is not supported by functional evidence. Instead, a population of PDGFR$\alpha^+$ interstitial cells might serve as 'negative pacemakers', providing a suppressive influence on DSMCs that prevents detrusor overexcitability and premature bladder emptying or sensations of fullness.

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

## Additional information

### Competing interests

The authors have no competing interests to declare.

### Author contributions

B.T.D. and C.S.G. conceived the article, prepared the initial draft, and prepared initial figures. N.G. and A.M. provided further additions to the manuscript after the initial draft. All authors edited the figures and manuscript for intellectual content. All authors have read and approved the final version of this manuscript and agree to be accountable for all aspects of the work in ensuring that questions related to the accuracy or integrity of any part of the work are appropriately investigated and resolved. All persons designated as authors qualify for authorship, and all those who qualify for authorship are listed.

## Funding

This work was supported by funding from the Irish Higher Education Authority Technological University Transformation Fund TUTFY2144 (to BTD) and TUTFY116 (to CSG).

## Acknowledgements

## Keywords

Ano1, asynchronous, bladder, calcium channels, continence, interstitial cell, oscillation, pacemaker, rhythmicity, smooth muscle, urethra

## Supporting information

Additional supporting information can be found online in the Supporting Information section at the end of the HTML view of the article. Supporting information files available:

**Peer Review History**

