## [Peer Review History · The Journal of Physiology]

Cells and ionic conductances contributing to spontaneous activity in bladder and urethral smooth muscle

Bernard Thomas Drumm, Neha Gupta, Alexandru Mircea, and Caoimhin Griffin

DOI: 10.1113/JP284744

Corresponding author(s): Bernard Drumm (bernard.drumm@dkit.ie)

Review Timeline:

Submission Date:	01-Dec-2023
Editorial Decision:	25-Jan-2024
Revision Received:	09-Aug-2024
Editorial Decision:	27-Aug-2024
Revision Received:	27-Aug-2024
Accepted:	02-Sep-2024

Senior Editor: *Laura Bennet*

Reviewing Editor: *T Alexander Quinn*

Transaction Report:

Dear Dr Drumm,

Re: JP-TR-2023-284744 "Does the lower urinary tract need a pacemaker? Origin and function of rhythmicity in the bladder and urethra" by Bernard Thomas Drumm, Neha Gupta, Alexandre Mircea, and Caoimhin Griffin

Thank you for submitting your manuscript to The Journal of Physiology. It has been assessed by a Reviewing Editor and by 1 expert referee and we are pleased to tell you that it is potentially acceptable for publication following satisfactory major revision.

LANGUAGE EDITING AND SUPPORT FOR PUBLICATION: If you would like help with English language editing, or other article preparation support, Wiley Editing Services offers expert help, including English Language Editing, as well as translation, manuscript formatting, and figure formatting at www.wileyauthors.com/eoo/preparation. You can also find resources for Preparing Your Article for general guidance about writing and preparing your manuscript at www.wileyauthors.com/eoo/prepresources.

REVISION CHECKLIST:

We look forward to receiving your revised submission.

Yours sincerely,

Professor Laura Bennet
Senior Editor
The Journal of Physiology
<https://jp.msubmit.net>
<http://jp.physoc.org>
The Physiological Society
Hodgkin Huxley House
30 Farringdon Lane
London, EC1R 3AW
UK
<http://www.physoc.org>
<http://journals.physoc.org>

REQUIRED ITEMS

- Please include an Abstract Figure file, as well as the Figure Legend text within the main article file. The Abstract Figure is a piece of artwork designed to give readers an immediate understanding of the Review Article and should summarise the main conclusions. If possible, the image should be easily 'readable' from left to right or top to bottom. It should show the physiological relevance of the Review so readers can assess the importance and content of the article. Abstract Figures should not merely recapitulate other figures in the Review. Please try to keep the diagram as simple as possible and without superfluous information that may distract from the main conclusion of the Review. Abstract Figures must be provided by authors no later than the revised manuscript stage and should be uploaded as a separate file during online submission labelled as File Type 'Abstract Figure'. Please ensure that you include the figure legend in the main article file. All Abstract Figures will be sent to a professional illustrator for redrawing and you may be asked to approve the redrawn figure before your paper is accepted.

- Your MS must include a complete "Additional information section" with the following 4 headings and content:

Competing Interests: A statement regarding competing interests. If there are no competing interests, a statement to this effect must be included. All authors should disclose any conflict of interest in accordance with journal policy.

Author contributions: Each author should take responsibility for a particular section of the study and have contributed to writing the paper. Acquisition of funding, administrative support or the collection of data alone does not justify authorship; these contributions to the study should be listed in the Acknowledgements. Additional information such as 'X and Y have contributed equally to this work' may be added as a footnote on the title page.

It must be stated that all authors approved the final version of the manuscript and that all persons designated as authors qualify for authorship, and all those who qualify for authorship are listed.

Funding: Authors must indicate all sources of funding, including grant numbers. If authors have not received funding, this must be stated.

It is the responsibility of authors funded by RCUK to adhere to their policy regarding funding sources and underlying research material. The policy requires funding information to be included within the acknowledgement section of a paper. Guidance on how to acknowledge funding information is provided by the Research Information Network. The policy also requires all research papers, if applicable, to include a statement on how any underlying research materials, such as data, samples or models, can be accessed. However, the policy does not require that the data must be made open. If there are considered to be good or compelling reasons to protect access to the data, for example commercial confidentiality or legitimate sensitivities around data derived from potentially identifiable human participants, these should be included in the statement.

Acknowledgements: Acknowledgements should be the minimum consistent with courtesy. The wording of acknowledgements of scientific assistance or advice must have been seen and approved by the persons concerned. This section should not include details of funding.

- Please upload separate high quality figure files via the submission form.

- Author profile(s) must be uploaded via the submission form. Authors should submit a short biography (no more than 100 words for one author or 150 words in total for two authors) and a portrait photograph of the two leading authors on the paper. These should be uploaded and clearly labelled together in a Word document with the revised version of the manuscript. Any standard image format for the photograph is acceptable, but the resolution should be at least 300 DPI and

preferably more. A group photograph of all authors is also acceptable, providing the biography for the whole group does not exceed 150 words.

- It is the authors' responsibility to obtain any necessary permissions to reproduce previously published material and to list these within the main article file. For information, please see: https://jp.msubmit.net/cgi-bin/main.plex?form_type=display_requirements#permissions.

EDITOR COMMENTS

Reviewing Editor:

Your review addressing the potential role of intrinsic pacemaking activity in the smooth muscle tissues of the urethra and bladder in the lower urinary tract has been reviewed by an expert in the field. They felt it has the potential to be influential, as the urethra has not been the subject of extensive research, so the summary of current work on the cellular physiology of the urethral smooth muscle layer is a useful contribution. They thought, though, that this could be enhanced by some focus on the translational potential of current research, to provide possible insight into mechanisms that might underlie common benign disorders of lower urinary function (such as detrusor overactivity and bladder outlet obstruction).

The review did, though, have some reservations about the article as currently presented, which need to be addressed. The main concern related to the difference between pacemaking activity and spontaneous activity, which they felt was not clear or well defined, and needs to be better addressed in the Introduction and adhered to throughout to help justify the suggestion that pacemaking may be an important aspect of lower urinary tract function. They were not convinced that pacemaking activity is the correct term, as it implies the initiation of a regular cycle of activity that propagates to other normally quiescent cells (such as occurs in the heart, or with cells in the renal pelvis that initiate peristaltic waves in the ureter), versus spontaneous activity, which may be generated in smooth muscle-lined tissues in a less co-ordinated fashion to set basal levels of wall stress (which seems to then be the more the desired attribute required of the bladder and, in particular, the urethral wall).

Along with this, they had additional questions and concerns about information in the paper and its interpretation, which are outlined in the Comments for the Author and should be addressed in a detailed response and revision of the manuscript.

REFEREE COMMENTS

Referee #1:

This submission addresses the question of whether the smooth muscle tissues of the lower urinary tract (urethra and bladder) possess intrinsic pacemaking activity to activate these tissues.

The major issue that arises is why these tissues require pacemaking activity, as one understands it, from the need to remove urine from the lower urinary tract or retain it in the bladder. Bladder contraction when activated by the autonomic nervous system is powerful enough to expel its contents into a receptive outflow tract and thus facilitate flow through the urethra. This might be contrasted to the ureter that is a low-pressure conductive tract that requires a true pacemaker in the renal pelvis to initiate directional peristaltic flow from the kidney to the bladder.

I think there is a confusion in identifying pacemaker activity with spontaneous activity. The latter may have a basic role in the lower urinary tract to set the longer-term control of wall stress (i.e. muscle tone) in the bladder and urethra.

There have been numerous studies and reviews on the topic of spontaneous activity, especially in the bladder wall, but less in the urethral wall, and this review provides some valuable comparison. However, I think, the basic premise of what is the relevant phenomenon (pacemaking or spontaneous activity) to be investigated needs to be established more convincingly in the Introduction. If the latter, generation of spontaneous activity, especially in the bladder, has been relatively well explored in the literature. This submission concentrates on a consideration of this phenomenon from its origins in the smooth muscle layers of the ureter and bladder walls.

General observations

1. The title of the submission is a challenging one ('Does the lower urinary tract require a pacemaker?') and in my view needs much more justification in the Introduction, see above. Pacemaker activity in smooth muscle-lined organs can be considered to drive peristalsis and co-ordinate directional flow of contents when there are only small overall pressure gradients, as in the ureter.

Does the ureter require pacemaking activity when the bladder provides a substantial pressure head to direct flow, when the outflow tract sphincters are relaxed?

Does the bladder require a pacemaker when the pelvic parasympathetic nerve reasonably quickly co-ordinates activation of the detrusor layer from the bladder neck throughout the dome to provide such a pressure head?

2 An equally important question, in the opinion of this referee, is the role of disco-ordinated spontaneous activity that will determine an increase or decrease of wall stress ('tone') of the bladder and urethra. Does this arise from specific pacemakers or from more stochastic generation of semi-coordinated local activity (especially as such activity is not confined to specific loci but emerges from a shifting landscape of localised activity)? Spontaneous activity could be facilitated and even initiated by many of the mechanisms discussed in this submission; but puts a different direction to the overall physiological processes under consideration. There is no mention of spontaneous activity despite the plethora of evidence for it, especially in the bladder - this comes from cystometry, isolated bladder/detrusor recordings and optical imaging studies. Furthermore, spontaneous activity becomes pathologically more evident in several conditions associated with benign lower urinary tract disorders, such as idiopathic and neurogenic detrusor overactivity.

3 It is implied on line 78 that over-excitability of detrusor smooth muscle can lead to lower urinary tract symptoms (LUTS) - this is rather over-simplistic. It might contribute to detrusor overactivity and several, not mutually exclusive, theories have been proposed, but considered here. Evidence for a 'myogenic' cause of DO is actually thin and other hypotheses have been investigated, such as neurogenic central and peripheral overactivity; and overstimulation of the so-called mucosal sensory web, which have equally convincing, if not better, experimental evidence. Perhaps the sensory web might be expanded upon where appropriate - i.e. how distortion of the mucosa during bladder filling may release neuroactive agents that may in turn affect local sensory afferents and even underlying smooth muscle both in the true detrusor layer and the muscularis mucosae of the lamina propria, and so provide local, peripheral nervous and central nervous responses.

4 There is mention of the urethral rhabdosphincter as a contributor to outflow tract resistance (para commencing line 103). However, its role seems to be a little downplayed, to one responsible for protecting continence during bladder filling with large changes to abdominal pressure. This is in contrast to EMG evidence for a key role immediately preceding voiding and in clinical conditions such as detrusor sphincter dyssynergia. Moreover EMG responses in the micturition cycle are different in several small laboratory animal and humans that may belie several functions of rhabdosphincter function. Moreover, pudendal nerve damage, which innervates the rhabdosphincter, can also have a profound effect on normal continence, which implies a more fundamental importance.

It may be worth mentioning here, that another contributor to maintenance of urethral resistance, is a urethral wall turgor pressure initiated by altering blood flow to the mucosal layer - Brading and Greenland.

5 Please check general spelling - for example 'prostrate' for 'prostate'.

Specific questions

1 Line 136. The bladder is not totally impervious to water and electrolyte transport, although it is very low; aquaporins, ion transporters and mineralocorticoid receptors are present in urothelial/mucosal layers.

2 Line 154. The possibility that there are innervated mechanoreceptors in the detrusor layer has never been described - despite such statements in some urology textbooks.

3 Line 158 et seq. The section is headed "How does rhythmic activity ... generate tone?" Here I see the concatenation between pacemaker activity and tone. One does not necessarily decide the other, are you speaking of spontaneous activity or pacemaking activity? In all there is no consideration of how 'activity' (presumably electrical) can spread throughout the tissue - vital for understanding how gross spontaneous or pacemaking activity can occur. Do the cable properties allow such spread? Are there gap junctions between relevant cells? - i.e. is there a relevant biophysical substrate to allow this to happen. Here the analogy with true GI-ICC (line 209) may or may not be relevant.

4 Line 256. The authors make the point that c-Kit labelling is unreliable between tissues, animals and lab groups. Is there any resolution, in their view, to this disparity.

5 Line 261 and next two paragraphs. The authors turn to the issue of information transfer between "interstitial cells" and either neural inputs or muscle cells. The situation in urethral cells remains unclear to me and seems different from the GI-tract where electrotonic coupling between ICC and muscle cells has been shown (e.g. GD Hirst). Some definitive statement would be useful at the end of this section about the evidence for urethral ICC coupling with other target cells.

6 Line 368. Non-voiding bladder contractions (NVCs) are introduced. It might be beneficial to distinguish these from low level spontaneous contractions as many use NVCs as evidence of pathological contractions. Optical imaging studies identify higher frequency spontaneous contractions in normal bladders, that occupy more discrete areas of the bladder wall, and those from pathological bladders that occupy greater areas but are less frequent and more associated with NVCs in bladder cystometry. Also please quote a paper to show NVCs are TTX-insensitive to give the word 'appears' less weight in the sentence on line 372.

7 Line 382 et seq. There is abundant evidence that spontaneous contractions are much more prevalent with an intact mucosa and suggests interaction between the mucosa and detrusor layers. Moreover, optical imaging studies show that Ca²⁺ and membrane potential signals appear in the mucosal layer first before propagating to the detrusor layer. How can

this be built into such a model?

8 Line 380. If stretch were to be a significant activator of detrusor spontaneous/pacemaking activity this would be apparent in the filling phase of the micturition cycle when by contrast bladder wall compliance is greatest and bladder wall stress low. It could be that a (an additional) function of Ca²⁺ channels is to maintain adequate filling of Ca-stores for subsequent muscarinic receptor activation of the detrusor layer. This would be consistent with a pathological role of such a mechanisms, as NVCs become more apparent and bladder compliance is decreased.

9 Line 442 et seq. The next two paragraphs are a comprehensive review of K⁺ conductances in detrusor. There is also evidence that application of muscarinic receptor agonists generates a hyperpolarisation along with a rise of intracellular Ca²⁺ that is consistent with activation of a Ca²⁺-activated K⁺ conductance on release of Ca²⁺ from intracellular Ca-stores.

10 Line 395. Could some information be provided about the Cx subtype between adjacent detrusor myocytes, i.e. Cx43 or Cx45.

11 Line 520 et seq. There is evidence that a subset of interstitial cells isolated from the lamina propria at least have an excitatory phenotype of a Ca²⁺-dependent Cl⁻ current, with little evidence of an inward Ca²⁺ current. How might this fit into the above scheme and the clear dependence of spontaneous activity on an intact mucosa?

12 Line 589 et seq. This represents an honest appraisal of the mechanisms that might underlie spontaneous activity in urethral and detrusor smooth muscle. An important consideration is species variability which suggests that spontaneous activity may have different roles in different species. Is there evidence that spontaneous activity has different characteristics in small animals (mice, rats, guinea-pigs) compared to those in larger animals (humans, sheep, pigs)?

END OF COMMENTS

Confidential Review

01-Dec-2023

Your review addressing the potential role of intrinsic pacemaking activity in the smooth muscle tissues of the urethra and bladder in the lower urinary tract has been reviewed by an expert in the field. They felt it has the potential to be influential, as the urethra has not been the subject of extensive research, so the summary of current work on the cellular physiology of the urethral smooth muscle layer is a useful contribution. They thought, though, that this could be enhanced by some focus on the translational potential of current research, to provide possible insight into mechanisms that might underlie common benign disorders of lower urinary function (such as detrusor overactivity and bladder outlet obstruction).

The review did, though, have some reservations about the article as currently presented, which need to be addressed. The main concern related to the difference between pacemaking activity and spontaneous activity, which they felt was not clear or well defined, and needs to be better addressed in the Introduction and adhered to throughout to help justify the suggestion that pacemaking may be an important aspect of lower urinary tract function. They were not convinced that pacemaking activity is the correct term, as it implies the initiation of a regular cycle of activity that propagates to other normally quiescent cells (such as occurs in the heart, or with cells in the renal pelvis that initiate peristaltic waves in the ureter), versus spontaneous activity, which may be generated in smooth muscle-lined tissues in a less coordinated fashion to set basal levels of wall stress (which seems to then be the more the desired attribute required of the bladder and, in particular, the urethral wall).

Along with this, they had additional questions and concerns about information in the paper and its interpretation, which are outlined in the Comments for the Author and should be addressed in a detailed response and revision of the manuscript.

Thank you to the reviewing editor for their work on our manuscript. We have reviewed the referee comments carefully and amended the paper accordingly. We address the specific points raised below in blue font.

REFEREE COMMENTS

Referee #1:

This submission addresses the question of whether the smooth muscle tissues of the lower urinary tract (urethra and bladder) possess intrinsic pacemaking activity to activate these tissues. The major issue that arises is why these tissues require pacemaking activity, as one understands it, from the need to remove urine from the lower urinary tract or retain it in the bladder. Bladder contraction when activated by the autonomic nervous system is powerful

enough to expel its contents into a receptive outflow tract and thus facilitate flow through the urethra. This might be contrasted to the ureter that is a low-pressure conductive tract that requires a true pacemaker in the renal pelvis to initiate directional peristaltic flow from the kidney to the bladder.

We would like to thank the reviewer for their careful review of our manuscript. The reviewer has pointed out an important issue that is essential for us to clarify. They are absolutely correct that contraction activated by the autonomic nervous system is sufficiently powerful to expel urine from the bladder. However, in our review, we focus entirely on the spontaneous activity (contractile, calcium signalling and electrical activity) that occurs in ex vivo detrusor and urethral samples during the filling phase. We have not described mechanisms that coordinate voiding beyond a few cursory statements in the introduction to provide some context for readers. This is a very important point to clarify and we apologize if this was not abundantly clear from the initial draft. We have made the focus of the review more explicit in the revised version.

I think there is a confusion in identifying pacemaker activity with spontaneous activity. The latter may have a basic role in the lower urinary tract to set the longer-term control of wall stress (i.e. muscle tone) in the bladder and urethra. There have been numerous studies and reviews on the topic of spontaneous activity, especially in the bladder wall, but less in the urethral wall, and this review provides some valuable comparison. However, I think, the basic premise of what is the relevant phenomenon (pacemaking or spontaneous activity) to be investigated needs to be established more convincingly in the Introduction. If the latter, generation of spontaneous activity, especially in the bladder, has been relatively well explored in the literature. This submission concentrates on a consideration of this phenomenon from its origins in the smooth muscle layers of the ureter and bladder walls.

We have striven in the revised manuscript to better clarify the focus of the review, which is the origin of spontaneous activity in these LUT organs. We also seek to address whether non-SMC contribute to this activity or if it is an intrinsic ability of SMC to generate this activity alone. The influence of non-SMC cell types has been referred to as a form of pacemaking in the literature, but we have now avoided this label in the revision to avoid confusion.

General observations

1. The title of the submission is a challenging one ('Does the lower urinary tract require a pacemaker?') and in my view needs much more justification in the Introduction, see above. Pacemaker activity in smooth muscle-lined organs can be considered to drive peristalsis and co-ordinate directional flow of contents when there are only small overall pressure gradients, as in the ureter.

Based on the reviewers comments we have amended the title of our review to "Cells and ionic conductances contributing to spontaneous activity in bladder and urethral smooth muscle". We

believe this new title better reflects the discussion points and focus of the paper. Thank you for the suggestion to reconsider this.

Does the ureter require pacemaking activity when the bladder provides a substantial pressure head to direct flow, when the outflow tract sphincters are relaxed?

The ureter is known to have non-SMC cell types, which are similar to pacemaker ICC in the gut. These cells express kit and are hypothesized to influence excitability of SMC in this tissue to coordinate peristaltic contractions to propel urine to the bladder for storage. We have not discussed the ureter in our review, as we have solely focused on the lower urinary tract. The existence and role of pacemaker cells in the upper urinary tract has been the subject of a recent extensive review here: Grainger N. Identifying peristaltic pacemaker cells in the upper urinary tract. *J Physiol*. 2024 Jan 5. doi: 10.1113/JP284754. Epub ahead of print. PMID: 38180778.

Does the bladder require a pacemaker when the pelvic parasympathetic nerve reasonably quickly co-ordinates activation of the detrusor layer from the bladder neck throughout the dome to provide such a pressure head?

We appreciate the reviewer's interest in the potential mechanisms that IC-like cells or pacemakers could mediate voiding contractions in the bladder. We also agree that this is an area that needs major exploration. Pelvic parasympathetic initiation of bladder voiding contractions occurs during micturition. However, our review focuses on reviewing the evidence behind the cellular mechanisms underlying spontaneous activity in bladder smooth muscle during the filling phase, in the absence of parasympathetic neural influence. Upon review of the literature, evidence suggests that a primary pacemaker doesn't exist in the bladder, however, emerging evidence proposes an interstitial-like cell present (not exclusively) in the detrusor, negatively dampens the excitability of bladder smooth muscle during the filling phase. Little is known about the role of PDGFR α cells during micturition. One study showed that the amplitude of the sustained component, not the initial peak, of EFS-evoked detrusor contractions in mice was enhanced when P2Y1 receptors were inhibited (Lee et al., 2014). P2Y receptors are expressed on PDGFR α cells and not DSMCs, suggesting that they could potentially regulate voiding contractions, at least in mice. Despite this study, considering that the purinergic component only contributes to around 5% of the voiding response in humans, one could easily speculate that PDGFR α cells wouldn't significantly influence human micturition events. However, these mechanisms and others need further elucidation.

Lee H, Koh BH, Peri LE, Sanders KM, Koh SD. Purinergic inhibitory regulation of murine detrusor muscles mediated by PDGFR α + interstitial cells. *J Physiol*. 2014 Mar 15;592(6):1283-93

2 An equally important question, in the opinion of this referee, is the role of dis-coordinated spontaneous activity that will determine an increase or decrease of wall stress ('tone') of the bladder and urethra. Does this arise from specific pacemakers or from more stochastic generation of semi-coordinated local activity (especially as such activity is not confined to specific loci but emerges from a shifting landscape of localized activity)? Spontaneous activity

could be facilitated and even initiated by many of the mechanisms discussed in this submission; but puts a different direction to the overall physiological processes under consideration. There is no mention of spontaneous activity despite the plethora of evidence for it, especially in the bladder - this comes from cystometry, isolated bladder/detrusor recordings and optical imaging studies. Furthermore, spontaneous activity becomes pathologically more evident in several conditions associated with benign lower urinary tract disorders, such as idiopathic and neurogenic detrusor overactivity.

We have now made spontaneous activity the focus of the review and it is mentioned throughout the manuscript, with some specific sections even referred to this directly in the headings ("*Spontaneous activity and tone in urethral smooth muscle*" and "*Spontaneous 'Rhythmical' activity in the bladder*"). In addition, we specifically dedicated a large section of the review to how disco-ordinated activity leads to an increase in tone or 'wall stress' as the reviewer puts it, in the urethra, as well as the bladder. Figure 2 in our manuscript, as well as its accompanying text, is in fact a detailed description of this phenomenon. As a result, we feel that this section has improved considerably.

3 It is implied on line 78 that over-excitability of detrusor smooth muscle can lead to lower urinary tract symptoms (LUTS) - this is rather over-simplistic. It might contribute to detrusor overactivity and several, not mutually exclusive, theories have been proposed, but considered here. Evidence for a 'myogenic' cause of DO is actually thin and other hypotheses have been investigated, such as neurogenic central and peripheral overactivity; and overstimulation of the so-called mucosal sensory web, which have equally convincing, if not better, experimental evidence. Perhaps the sensory web might be expanded upon where appropriate - i.e. how distortion of the mucosa during bladder filling may release neuroactive agents that may in turn affect local sensory afferents and even underlying smooth muscle both in the true detrusor layer and the muscularis mucosae of the lamina propria, and so provide local, peripheral nervous and central nervous responses.

We agree with the reviewer's thoughts on the critical role of the mucosa and mucosal sensory web in regulating the excitability of the underlying detrusor layer, in addition to the mechanisms underlying the pathophysiological causes of LUTs (i.e. Myogenic, Neurogenic, & Sensory/peripheral theories). This review concentrates on the cells and ionic conductances present in the detrusor layer, and therefore we are restricted in the scope and detail we can provide on the contribution of the mucosa to spontaneous activity, due to word count limitations associated with this submission. As such, we constrained this review to comparing and contrasting information regarding the smooth muscle layers in both the bladder and urethra.

4 There is mention of the urethral rhabdosphincter as a contributor to outflow tract resistance (para commencing line 103). However, its role seems to be a little downplayed, to one responsible for protecting continence during bladder filling with large changes to abdominal pressure. This is in contrast to EMG evidence for a key role immediately preceding voiding and

in clinical conditions such as detrusor sphincter dyssynergia. Moreover EMG responses in the micturition cycle are different in several small laboratory animal and humans that may belie several functions of rhabdosphincter function. Moreover, pudendal nerve damage, which innervates the rhabdosphincter, can also have a profound effect on normal continence, which implies a more fundamental importance.

There is an increased awareness clinically of the importance of the smooth muscle layers of the urethra in maintaining continence and that a historical emphasis on the striated muscle layer has not led to the development of adequate non-surgical means of alleviating urethral disorders. The focus of our review is on bladder and urethral smooth muscle; thus we feel that only a cursory description of the role of urethral striated muscle is warranted, along with a brief description of the key studies that outline the importance of the smooth muscle layers to generating tone and contributing to urethral closure pressure. These are cited in our introduction.

For a recent update on these findings, see here:

Venema PL, Kramer G, van Koevinge GA, Heesakkers JPFA. The Maximal Urethral Pressure at Rest and during Normal Bladder Filling Is Only Determined by the Activity of the Urethral Smooth Musculature in the Female. *J Clin Med.* 2023 Mar 29;12(7):2575. doi: 10.3390/jcm12072575. PMID: 37048657; PMCID: PMC10095129.

It may be worth mentioning here, that another contributor to maintenance of urethral resistance, is a urethral wall turgor pressure initiated by altering blood flow to the mucosal layer - Brading and Greenland.

Thank you for this suggestion, we have added the following text to the amended introduction: | The urethral vasculature may also contribute to tone and urethral resistance pressure by applying a passive turgor pressure initiated by altering blood flow to the mucosal layer (Greenland & Brading, 1997; Hashitani et al., 2024a).

5 Please check general spelling - for example 'prostrate' for 'prostate'.

Thank you, now amended.

Specific questions

1 Line 136. The bladder is not totally impervious to water and electrolyte transport, although it is very low; aquaporins, ion transporters and mineralocorticoid receptors are present in urothelial/mucosal layers.

In the revised manuscript, Line 136 has been changed to “The urothelium, the innermost layer of the bladder, plays a critical role in maintaining water tightness. While not entirely impermeable, it acts as a highly effective barrier, allowing for a very slow passage of water and electrolytes.”

2 Line 154. The possibility that there are innervated mechanoreceptors in the detrusor layer has never been described - despite such statements in some urology textbooks.

We appreciate the reviewers' comments highlighting the lack of knowledge regarding mechanoreceptor expression/function on detrusor layer afferent nerves. We have significantly amended line 154 to avoid such confusion.

3 Line 158 et seq. The section is headed "How does rhythmic activitygenerate tone?" Here I see the concatenation between pacemaker activity and tone. One does not necessarily decide the other, are you speaking of spontaneous activity or pacemaking activity? In all there is no consideration of how 'activity' (presumably electrical) can spread throughout the tissue - vital for understanding how gross spontaneous or pacemaking activity can occur. Do the cable properties allow such spread? Are there gap junctions between relevant cells? - i.e. is there a relevant biophysical substrate to allow this to happen. Here the analogy with true GI-ICC (line 209) may or may not be relevant.

Based on the reviewers comments, we have carefully amended this section to make clear that we are speaking about how spontaneous activity in USMC is generated, and whether any non-SMC cells influence this. We have explicitly avoided the use of the term 'pacemaker' here to avoid confusion in the revised paper. We describe at length in this section how USMC are coupled, detailing the connexin sub types known to be present. In the urethra, while ICC-LC and USMC express gap junctions, specifically connexin 43 and 37 in rat and sheep urethra (Sancho et al., 2011), this coupling is relatively poor. We further describe the concept of 'local control' of USMC bundle contractions and how in the mouse at least, voltage dependent mechanisms are lacking and not required for USMC spontaneous activity. We also further discuss in later sections how biophysical substrates and molecules between interstitial cells and USMC might allow for non-electrical forms of communication based on findings in other cell types in other organs.

4 Line 256. The authors make the point that c-Kit labelling is unreliable between tissues, animals and lab groups. Is there any resolution, in their view, to this disparity.

Yes actually there is a recent technical advancement that has helped to alleviate this. The following text is now included in the revised manuscript. “Mice expressing the genetically encoded Ca²⁺ indicator, GCaMP, expressed exclusively in Kit⁺ cells with Cre recombinase driven from Kit (c-Kit⁺/Cre-ERT2) has allowed investigators to distinguish ICC-LC from SMC in tissues where Kit antibody labelling of ICC-LC has proven challenging, such as the mouse

urethra (Gupta et al., 2024) and renal pelvis (Grainger et al., 2020). Labelling mouse Kit-GCaMP tissues with antibodies against green fluorescent protein (GFP), enables Kit⁺ ICC-LC expressing GCaMP to be visualized in immunohistochemical imaging, owing to antibody labelling of the GFP molecule which is a constitute component of GCaMP.”

5 Line 261 and next two paragraphs. The authors turn to the issue of information transfer between "interstitial cells" and either neural inputs or muscle cells. The situation in urethral cells remains unclear to me and seems different from the GI-tract where electrotonic coupling between ICC and muscle cells has been shown (e.g. GD Hirst). Some definitive statement would be useful at the end of this section about the evidence for urethral ICC coupling with other target cells.

As we outline in the review, how ICC-LC might couple to USMC appears to be species dependent. In rabbit, there may be a similar connection between ICC-LC and USMC and intramuscular ICC (ICC-IM) and GI SMC, where stochastic activation of Ano1 in ICC-IM increases the overall excitability of SMC. However, in the urethra, due to the noted poor electrical coupling of the tissue, how strong this association is in the rabbit is unclear. In the mouse, due to the lack of a voltage-dependent mechanism to influence USMC behaviours, how ICC-LC activity might couple to USMC is presently unknown. We have added 3 detailed paragraphs into the revised manuscript on this point. While there is no direct evidence in the mouse urethra, we proposed several lines of possible non-electrical communication from other organ systems that might account for how ICC-LC might influence USMC, although this needs to be experimentally examined in future work.

6 Line 368. Non-voiding bladder contractions (NVCs) are introduced. It might be beneficial to distinguish these from low level spontaneous contractions as many use NVCs as evidence of pathological contractions. Optical imaging studies identify higher frequency spontaneous contractions in normal bladders, that occupy more discrete areas of the bladder wall, and those from pathological bladders that occupy greater areas but are less frequent and more associated with NVCs in bladder cystometry. Also please quote a paper to show NVCs are TTX-insensitive to give the word 'appears' less weight in the sentence on line 372.

We thank the reviewer for highlighting potential confusion between NVCs and spontaneous contractions. A description of the difference between NVCs and spontaneous contractions has been added to the revised manuscript. Additionally, an appropriate reference showing that rat NVCs are insensitive to TTX has been added (Kanai et al., 2007).

7 Line 382 et seq. There is abundant evidence that spontaneous contractions are much more prevalent with an intact mucosa and suggests interaction between the mucosa and detrusor layers. Moreover, optical imaging studies show that Ca²⁺ and membrane potential signals

appear in the mucosal layer first before propagating to the detrusor layer. How can this be built into such a model?

We agree with the reviewer that strong evidence exists showing that mucosa-intact bladder strips exhibit greater spontaneous activity, showing that the mucosa greatly modulates spontaneous activity of the bladder. This is exemplified by the study by Kanai et al., 2007 which shows that mechanical stimulation of the mucosa or application of low concentrations of Carbachol (50 nM) can initiate mucosa-localized Ca^{2+} waves and membrane depolarizing signals that propagate throughout the urothelial-suburothelial regions, before propagating in the transverse direction towards the detrusor layer. However, this study also shows that Ca^{2+} events can initiate in the detrusor layer itself, specifically the serosal end of the muscle layer. Indeed, an abundance of evidence exists showing that denuded mucosa-free detrusor strips exhibit robust spontaneous rhythmic contractions. Therefore, it is clear that spontaneous activity can originate in the detrusor layer irrespective of the influence of the mucosa. This review manuscript exclusively concentrates on spontaneous activity originating in the detrusor smooth muscle.

8 Line 380. If stretch were to be a significant activator of detrusor spontaneous/pacemaking activity this would be apparent in the filling phase of the micturition cycle when by contrast bladder wall compliance is greatest and bladder wall stress low. It could be that a (an additional) function of Ca^{2+} channels is to maintain adequate filling of Ca -stores for subsequent muscarinic receptor activation of the detrusor layer. This would be consistent with a pathological role of such a mechanisms, as NVCs become more apparent and bladder compliance is decreased.

In both isolated bladder myocytes and mucosa-free bladder sheet preparations, mechanical elongation of the myocytes or tissue induces propagating Ca^{2+} waves. Indeed, many cystometry studies have shown that as the bladder fills and as a result stretches, NVCs begin to develop in the bladder (Heppner et al., 2016). Isolated DMSCs display non-selective cation currents upon stretching of the membrane, resulting in depolarization of membrane potential and the development of an L-Type Ca^{2+} channel dependent action potential. L-Type channel-mediated action potentials underly spontaneous phasic activity of the bladder. Additionally, unpublished data from our lab has found that spontaneous activity of mucosa-free detrusor strips is greatly enhanced when the mechanosensitive ion channel, Piezo1, is activated. These findings, and others, strongly support the model that upon stretch, mechanosensitive ion channels on DSMCs activate leading to generation of and/or enhancement of L-type potential action potentials, and thus spontaneous phasic activity of the bladder. However, this model is based on spontaneous activity exclusively in mucosa-free detrusor strips. A more complicated model would take into account the significant influence of the mucosa on this activity.

The underlying Ca^{2+} events mediating detrusor spontaneous activity is reliant on L-type Ca^{2+} influx. Inhibitors of Ca^{2+} store refilling and or Ca^{2+} release (CPA, 2-APB, & ryanodine) enhance detrusor spontaneous activity (Hashitani et al., 2003; Herrera et al., 2000). This is believed to be

a result of reduced Ca^{2+} spark-BK channel activity, and important regulator of detrusor excitability during the filling phase. As far as we know, the role of L-type channels in refilling Ca^{2+} stores during spontaneous contractile activity of the detrusor are unknown. This pathway is difficult to study since phasic spontaneous contractions of the detrusor are completely reliant on L-type channel activity i.e. inhibition of L-type channels will result in no spontaneous detrusor contractions. We agree with the reviewer in their thoughts that store-refilling by L-type channels could be important to maintain adequate Ca^{2+} store load before Muscarinic-stimulation during a voiding contraction (Wu et al., 2002), since there is low level Ca^{2+} store release via Ca^{2+} sparks in DSMCs. Indeed, there is an abundance of evidence showing that voiding contractions rely, at least in part, on store release. However, with this manuscript we are focusing on spontaneous activity during the filling phase of the micturition cycle, the mechanisms involved nerve-mediated Muscarinic and Purinergic voiding contractions are extensive and very well established, it would be a review in itself.

9 Line 442 et seq. The next two paragraphs are a comprehensive review of K^{+} conductances in detrusor. There is also evidence that application of muscarinic receptor agonists generates a hyperpolarisation along with a rise of intracellular Ca^{2+} that is consistent with activation of a Ca^{2+} -activated K^{+} conductance on release of Ca^{2+} from intracellular Ca-stores.

We agree with the reviewer's comments that there is evidence of Ca^{2+} activated K^{+} channel activation, and subsequent mediation of the magnitude of muscarinic and purinergic-mediated responses of the bladder. However, these responses represent voiding contractions, whereas this review and the section describing the evidence of K^{+} channel conductances focuses on spontaneous detrusor contractions during the filling phase.

10 Line 395. Could some information be provided about the Cx subtype between adjacent detrusor myocytes, i.e. Cx43 or Cx45.

We thank the reviewer for their suggestions. Our revised manuscript now contains information on the Connexin proteins (Cx45 & Cx43) involved in electrical propagation across detrusor SMCs.

11 Line 520 et seq. There is evidence that a subset of interstitial cells isolated from the lamina propria at least have an excitatory phenotype of a Ca^{2+} -dependent Cl^{-} current, with little evidence of an inward Ca^{2+} current. How might this fit into the above scheme and the clear dependence of spontaneous activity on an intact mucosa?

We thank the reviewer for highlighting important evidence regarding interstitial cells, and we agree that the study mentioned (Wu et al., 2004), is important to highlight when describing evidence of Cl^{-} conductances in regulating detrusor contractility. We have added a description

of this study to the “Evidence of interstitial cell-mediated pacemaking in the bladder” section where we describe Cl⁻ currents.

In regard to the proposition that this subpopulation of cells serve as pacemakers that regulate detrusor contractility and how they might fit into a scheme, our thoughts are as follows; Overall evidence proposing that there is a primary pacemaker in the bladder, is unconvincing thus far. The fact that this sub-population of cells possess a chloride conductance which could potentially propagate a depolarizing signal to the underlying detrusor layer, or cause release of an excitatory chemical stimuli, is mitigated by the fact that Niflumic Acid has little effect on the spontaneous activity of mucosa-intact bladder strips taken from mature rats (Lam et al., 2014). Alternatively however, these cells could act as “middlemen” between urothelial cell ATP release and activation of afferent sensory nerves in the bladder, as suggested by the original paper by Wu et al. 2004. One potential avenue to further explore the role of ANO1 channels in mediating spontaneous detrusor activity is through use of better more specific pharmacological agents. For example, Niflumic acid which is used in the studies mentioned and others (Bijos et al., 2014), has a vast array of non-specific effects, notably blockade of L-type Ca²⁺ channels. Newer generation ANO1 blockers such as ANI9 would be a better tool to utilize in determining the role of ANO1 in regulating spontaneous activity. These experiments may unearth findings that could help clarify the role of ANO1 in IC-like cells and spontaneous activity.

12 Line 589 et seq. This represents an honest appraisal of the mechanisms that might underlie spontaneous activity in urethral and detrusor smooth muscle. An important consideration is species variability which suggests that spontaneous activity may have different roles in different species. Is there evidence that spontaneous activity has different characteristics in small animals (mice, rats, guinea-pigs) compared to those in larger animals (humans, sheep, pigs)?

Currently, there is a paucity of detailed comparative studies on the spontaneous activity at the cellular level in either organ between small animals and larger mammals. While some comparisons can be made, such as the occurrence of similar calcium signals and electrical behaviours in some instances (such as calcium flashes in detrusor which occur in mice and humans) detailed quantitative comparisons are lacking and worthy of further investigation.

Dear Ben,

Re: JP-TR-2024-284744R1 "Cells and ionic conductances contributing to spontaneous activity in bladder and urethral smooth muscle" by Bernard Thomas Drumm, Neha Gupta, Alexandru Mircea, and Caoimhin Griffin

Thank you for submitting your manuscript to The Journal of Physiology. It has been assessed by a Reviewing Editor and by 1 expert referee and we are pleased to tell you that it is acceptable for publication following satisfactory minor revision.

ABSTRACT FIGURES: Authors may use The Journal's premium BioRender account to create/redraw their Abstract Figures (and any other suitable schematic figure). Information on how to access this account is here: <https://physoc.onlinelibrary.wiley.com/journal/14697793/biorender-access>.

REVISION CHECKLIST: Upload a full Response to Referees file. To create your 'Response to Referees' copy all the reports, including any comments from the Senior and Reviewing Editors, into a Microsoft Word, or similar, file and respond to each point, using font or background colour to distinguish comments and responses and upload as the required file type.

We look forward to receiving your revised submission.

Best wishes,

Laura Bennet
Senior Editor

EDITOR COMMENTS

Reviewing Editor:

The reviewer is satisfied with the authors' response, but has asked for a couple of minor additional changes to the text for clarity, as outlined in the Comments for the Author. Please consider these additional requests in a final revision of the manuscript.

REFEREE COMMENTS

Referee #1:

The authors have carefully considered the questions raised by the reviewer.

I have no further comments, except a couple of very small points for clarity that may possibly be dealt with the editorial office.

1 Abstract, First sentence,

Lines 33,34

"Smooth muscle organs of the lower urinary tract comprise the bladder detrusor and the internal urethral sphincter, ..."

To many people, including many clinicians, the internal urethral sphincter represents the smooth muscle of the bladder neck. However, most of the work described here will likely be derived from experiments on smooth muscle derived from more downstream portions of the urethra.

May I suggest "Smooth muscle tissues of the lower urinary tract include the bladder detrusor and the urethral wall, ..."

2 Bladder anatomy and function

Lines 147-149

"Although the majority of afferent nerve fibres are located in the detrusor muscle layer.....",

The rest of sentence describes the principal role of mucosal afferents. There is no reference to this majority of fibres in the detrusor layer. The excellent work of Giorgio Gabella may answer the question.

I would suggest re-wording this sentence to "Afferent nerve endings are found in the detrusor and mucosal layers, but accumulating evidence suggests that the urothelium primarily dictates the activation of afferent nerves".

END OF COMMENTS

1st Confidential Review

09-Aug-2024

EDITOR COMMENTS

Reviewing Editor:

The reviewer is satisfied with the authors' response, but has asked for a couple of minor additional changes to the text for clarity, as outlined in the Comments for the Author. Please consider these additional requests in a final revision of the manuscript.

We have made the requested final changes. Thank you for your work on our MS.

REFEREE COMMENTS

Referee #1:

The authors have carefully considered the questions raised by the reviewer.

I have no further comments, except a couple of very small points for clarity that may possibly be dealt with the editorial office.

Thank you to the reviewer for their time and careful review of our MS. We have made the final requested changes as detailed below.

1 Abstract, First sentence,

Lines 33,34

"Smooth muscle organs of the lower urinary tract comprise the bladder detrusor and the internal urethral sphincter, ..."

To many people, including many clinicians, the internal urethral sphincter represents the smooth muscle of the bladder neck. However, most of the work described here will likely be derived from experiments on smooth muscle derived from more downstream portions of the urethra.

May I suggest "Smooth muscle tissues of the lower urinary tract include the bladder detrusor and the urethral wall, ..."

We have made the suggested change to this sentence.

2 Bladder anatomy and function

Lines 147-149

"Although the majority of afferent nerve fibres are located in the detrusor muscle layer.....",

The rest of sentence describes the principal role of mucosal afferents. There is no reference to this majority of fibres in the detrusor layer. The excellent work of Giorgio Gabella may answer the question.

I would suggest re-wording this sentence to "Afferent nerve endings are found in the detrusor and mucosal layers, but accumulating evidence suggests that the urothelium primarily dictates the activation of afferent nerves".

We have made the suggested change to this sentence.

Dear Ben,

Re: JP-TR-2024-284744R2 "Cells and ionic conductances contributing to spontaneous activity in bladder and urethral smooth muscle" by Bernard Thomas Drumm, Neha Gupta, Alexandru Mircea, and Caoimhin Griffin

We are pleased to tell you that your paper has been accepted for publication in The Journal of Physiology.

Authors should note that it is too late at this point to offer corrections prior to proofing. Major corrections at proof stage, such as changes to figures, will be referred to the Editors for approval before they can be incorporated. Only minor changes, such as to style and consistency, should be made at proof stage. Changes that need to be made after proof stage will usually require a formal correction notice.

Best wishes,

Laura Bennet
Senior Editor
The Journal of Physiology

P.S. - You can help your research get the attention it deserves! Check out Wiley's free Promotion Guide for best-practice recommendations for promoting your work at www.wileyauthors.com/eeo/guide. You can learn more about Wiley Editing Services which offers professional video, design, and writing services to create shareable video abstracts, infographics, conference posters, lay summaries, and research news stories for your research at www.wileyauthors.com/eeo/promotion.

IMPORTANT NOTICE ABOUT OPEN ACCESS: To assist authors whose funding agencies mandate public access to published research findings sooner than 12 months after publication, The Journal of Physiology allows authors to pay an Open Access (OA) fee to have their papers made freely available immediately on publication.

You can check if your funder or institution has a Wiley Open Access Account here: <https://authorservices.wiley.com/author-resources/Journal-Authors/licensing-and-open-access/open-access/author-compliance-tool.html>.

EDITOR COMMENTS

The authors have addressed all reviewer concerns and the paper is now acceptable for publication.